# Semi-continuum modelling of unsaturated porous media flow to explain the Bauters' paradox

Jakub Kmec[1], Miloslav Šír[1], Tomáš Fürst[2], and Rostislav Vodák[2]

[1]Palacký University Olomouc, Faculty of Science, Joint Laboratory of Optics of Palacký University and Institute of Physics of the Czech Academy of Sciences, 17. listopadu 1192/12, 779 00 Olomouc, Czech Republic
[2]Palacký University in Olomouc, Dpt. Mathematical Analysis and Applications of Mathematics, Faculty of Science, Olomouc, 771 46, Czech Republic

**Correspondence:** Jakub Kmec (jakub.kmec@upol.cz)

**Abstract.** In gravity-driven free infiltration of a wetting liquid into a homogeneous unsaturated porous medium, the flow pattern is known to depend significantly on the initial saturation. Point-source infiltration of a liquid into an initially dry porous medium produces a single finger with an oversaturated tip and an undersaturated tail. In an initially wet medium, a diffusion-like plume is produced with a monotonic saturation profile. We present a semi-continuum model based on a proper scaling of the retention curve which is discrete in space and continuous in time. We show that the semi-continuum model is able to describe this transition and to capture the experimentally observed dependence of the saturation overshoot and the finger velocity on the initial saturation.

## 1 Introduction

Infiltration of rainwater into soil forms an essential part of the hydrological cycle. Therefore, research on the movement of water in soil has long been a focus of attention. The origins of infiltration research were substantially influenced by the idea to describe the movement of water in soil by diffusion-like models (Richards, 1931). Later, it was discovered that – even in homogeneous porous materials – flow may become spatially very inhomogeneous. Most of the infiltrating water flows through preferential pathways leaving islands of dry material behind (Glass et al., 1988). This type of flow is well described by a semi-continuum model introduced in Vodák et al. (2022). In this paper, we demonstrate that this semi-continuum model captures infiltration into an unsaturated homogeneous porous medium comprehensively, in the sense that it correctly describes the experimentally observed complicated transition between the preferential and diffusion-like flow regimes.

There are three types of preferential flow (Nimmo, 2021): Macropore flow, funnel flow, and finger flow. Macropore flow proceeds through individual large pores that are highly conductive. The funnel flow is the result of heterogeneity of soil or rock hydraulic properties. Both macropore flow and funnel flow are features of non-homogeneous porous media. However, preferential flow (also known as finger flow) also occurs in a homogeneous medium and is believed to be caused by the wetting front instability (Glass et al., 1989a; Bauters et al., 2000; Sililo and Tellam, 2005; Aminzadeh and DiCarlo, 2010; DiCarlo, 2013; Wei et al., 2014; Cremer et al., 2017; Pales et al., 2018). A finger consists of two parts: an oversaturated *finger tip*

followed by a less saturated *finger tail*. It is widely accepted that this non-monotonicity of the saturation profile (i.e. saturation overshoot) is a necessary and sufficient condition of finger flow (DiCarlo, 2004; Rezanezhad et al., 2006).

It was experimentally observed that the magnitude of the saturation overshoot (i.e. the saturation difference between the finger tip and tail) depends on the infiltration rate (DiCarlo, 2004). For low influx, a stable wetting front forms without a saturation overshoot. For a larger flux, the saturation overshoot appears, and its magnitude increases with increasing flux up to a certain point beyond which the magnitude of the overshoot decreases again. For high flow rates, the saturation overshoot disappears completely. There is also a strong dependence of the flow regime on the initial saturation of the porous medium

(Bauters et al., 2000). For an initially dry medium, finger flow accompanied by saturation overshoot is observed. However, at a sufficiently high initial saturation (close to the saturated moisture limit), fingers do not form and a stable front dominates with no saturation overshoot (see Fig. 3 in Bauters et al. (2000)). Moreover, a non-monotonic dependence of wetting front velocity and finger width on the initial saturation was reported. With increasing initial saturation, the fingers first become more narrow and faster, but further increase in initial saturation makes them slow down, become thicker and more irregular, and gradually

disappear completely, giving way to diffusion-like plumes with no saturation overshoot. This is counterintuitive because one would expect the finger velocity to increase with increasing initial saturation. We call this complicated transition from finger-like regime to diffusion-like regime the *Bauters' paradox*, honoring the first author of the seminal article Bauters et al. (2000). Note that the preferential flow occurs also in highly saturated porous medium that is super-hydrophilic (Chen et al., 2022b). This complex behavior is not consistent with the standard theory which (1) does not allow for saturation overshoot behavior

(Fürst et al., 2009), (2) predicts an increase in wetting front velocity with increasing initial saturation (Bear, 1972), and (3) does not allow for preferential flow in a homogeneous medium.

    The standard model for unsaturated porous media flow is the Richards' Equation (RE) (Richards, 1931). RE is a combination of a mass balance equation and the Darcy-Buckingham law (Bear, 1972). It was shown by means of a mathematical proof that in the case of a constant influx into an initially dry homogeneous porous medium, RE is incompatible with saturation overshoot

because the RE is unconditionally stable (Fürst et al., 2009). The solution of the RE is stable in this case regardless of whether hysteresis of the retention curve is included because the hysteresis of the retention curve never comes into action. Thus, RE is not able to capture finger flow. There have been many attempts to model the flow in porous media differently; in principle, these attempts can be divided into continuum models (Hassanizadeh et al., 2002; Eliassi and Glass, 2002; Schneider et al., 2017; Brindt and Wallach, 2020; Beljadid et al., 2020; Cueto-Felgueroso et al., 2020; Ommi et al., 2022a, b), and pore-scale (discrete)

models (Lenormand et al., 1988; Blunt and Scher, 1995; Primkulov et al., 2018, 2019; Wei et al., 2022). Another approach is to combine the advantages of continuous and discrete modelling (Glass and Yarrington, 1989, 2003; Liu et al., 2005; Liu, 2017). Liu et al. (2005) and Liu (2017) developed an active region model in which fractal flow patterns are incorporated into the continuum approach. Glass and Yarrington (1989, 2003) proposed a unique Macro Modified Invasion Percolation model (MMIP), which is – in a single framework – able to capture finger flow, buoyancy-driven migration of gases through

porous media, and rough surface flow. However, saturation is not treated as a continuous quantity in the MMIP model, thus the model cannot reproduce the saturation overshoot or its dependence on initial saturation or influx rate. For a detailed review

of the different types of modelling, see e.g. Rooij (2000); DiCarlo (2010); Xiong (2014); Hunt and Sahimi (2017); Chen et al. (2022a).

Another attempt is reported in Kmec et al. (2019, 2021), who advocate for the so-called semi-continuum approach. In this approach, the porous medium is divided into a grid of blocks which are not considered infinitesimal – each block retains the nature of a porous medium and it is characterized by its pressure–saturation relation, hydraulic conductivity, and porosity. Saturation is considered continuous in time but constant throughout each block (i.e. piecewise constant in space). Flow between neighboring blocks proceeds according to the Darcy-Buckingham law. The key feature of the semi-continuum approach is to account for the block size. This is done by an appropriate scaling of the retention curve with the block size (Vodák et al., 2022). As the block size decreases, the retention curve becomes more flat (i.e. its derivative decreases) while keeping the hysteresis effect constant. See Vodák et al. (2022) for more details and a physical justification.

The semi-continuum model was shown to reproduce well all experimentally observed features of unsaturated porous medium flow in a long vertical tube (Kmec et al., 2019). A two dimensional version of the model was shown to correctly capture the transition between finger flow and diffusion-like flow with increasing initial saturation (Kmec et al., 2021) for uniform infiltration imposed on the entire top boundary. Vodák et al. (2022) examined the limit of the semi-continuum model as the block sizes go to zero. They report a limit version of the model in the form of a partial differential equation with a Prandtl-type hysteresis operator (Visintin, 1993) under the derivative.

In this paper, we use a previously developed semi-continuum model and demonstrate that this model is able to fully reproduce the Bauters' paradox – the transition from finger-like flow in initially dry medium to diffusion-like flow in initially wet medium for a point source infiltration. We show that the non-monotonic relation between initial saturation and flow velocity, and initial saturation and saturation overshoot magnitude is captured correctly by the semi-continuum model.

## 1.1 Bauters' paradox

The authors of Bauters et al. (2000) used a Hele-Shaw cell ($50 \times 30 \times 0.94$ cm) filled with homogeneous 20/30 quartz sand with particle size between $0.60$ mm and $0.85$ mm. The used sand does not change its wettability according to the duration of contact with distilled water. Water was injected at a rate of $2$ cm$^3$ min$^{-1}$ through a hypodermic needle located at the top of the chamber near the sand surface. The initial saturation was gradually increased from zero to the full field capacity. The results of the experiments can be summarized as follows:

– *Wetting front dependence on the initial saturation.* As the initial saturation increases, the flow regime changes from an unstable finger-like to a stable diffusion-like flow. Three flow regimes can be distinguished: unstable, intermediate, and stable. During unstable flow, the finger width remains almost constant. This is consistent with theoretical analysis (Raats, 1973) and experimental observations (Selker et al., 1992; Rezanezhad et al., 2006). In the intermediate regime, the fingers gradually give way to a stable infiltration front. This type of flow transition has not yet been sufficiently investigated, either theoretically or experimentally. In the diffusion-like regime, the saturation and pressure profiles are monotonic with no overshoot behavior. Moreover, the wetting front is much wider than the finger.

– *The width and velocity of the fingers.* With increasing initial saturation, the fingers first become faster and narrower, then they get slower and wider.

    – *Pressure and saturation overshoot magnitude.* The magnitude of the saturation overshoot decreases with increasing initial saturation of the medium. Moreover, a hyperbolic relation between initial saturation and saturation overshoot magnitude is observed. The same holds for the pressure overshoot magnitude.

Although the experiments of Bauters et al. (2000) are well known in the soil science community (currently more than 90 citations in the Scopus database), there is no unified explanation for the observed paradox. Moreover, almost none of the citing authors comment on this interesting phenomenon. To our best knowledge, there are only three attempts to model or explain Bauters' paradox. Chapwanya and Stockie (2010) used a dynamic capillary pressure term to model the effect of initial saturation. However, a small artificial perturbation in the influx had to be used to initiate the finger flow, and the influx was

imposed over the entire top boundary instead of a point. The finger velocity was independent of the initial saturation of the medium. Moreover, the authors did not focus on the non-monotonic dependence of finger width on initial saturation.

    Another attempt was undertaken by Joekar-Niasar and Hassanizadeh (2012) and Masoodi and Pillai (2012). The authors hypothesized that the non-monotonic velocity of the front is due to a trade-off between conductivity and capillary pressure. With increasing initial saturation, the conductivity increases because there is more trapped air in the medium. Beyond a critical

value of initial saturation, the trapping does not change significantly, but the matric potential decreases. As a result, the wetting front slows down. This means that the intrinsic permeability of the medium is not a constant but a function of saturation.

    Finally, Kmec et al. (2021) used a semi-continuum approach to investigate the effect of the initial saturation on the wetting flow formation. Similar to Chapwanya and Stockie (2010), the influx was imposed on the entire top boundary. The nonlinear dependence of the finger width was reproduced (see Fig. 6 in Kmec et al. (2021)). The finger velocity dependence on the initial

saturation was not studied due to different choice of the top boundary condition than in Bauters et al. (2000).

    This article presents simulations of the point source infiltration used in Bauters et al. (2000) by means of the semi-continuum model. We show that all the experimentally observed features of the Bauters' paradox are reproduced well.

## 2   Methods

### 2.1   Semi-continuum model

Let us recall the 2D semi-continuum model that was introduced by Kmec et al. (2021). Here, we use the same model with an appropriate scaling of the retention curve with the block size (Vodák et al., 2022). The porous medium is represented as a rectangular grid of $N \times M$ square blocks of uniform size $\Delta x \times \Delta x$. Each block is denoted by its row and column indices $[i, j]$. Saturation $S_t(i, j)$ and pressure $P_t(i, j)$ of the wetting phase (liquid) at time $t$ are assumed constant within each block $(i, j)$, and the pressure of the non-wetting phase (gas) is assumed to be zero everywhere. Each block retains the nature of a

porous medium and it is characterized by a hysteretic pressure–saturation relation (main wetting branch $p^w(S)$, main draining branch $p^d(S)$), non-hysteretic hydraulic conductivity (intrinsic permeability $\kappa$, relative permeability $k(S)$), and porosity $\theta$.

The invading wetting liquid is characterized by its density $\rho$ and dynamic viscosity $\mu$. Acceleration due to gravity is denoted by $g$. The semi-continuum model tracks the following three key quantities: The saturation $S_t(i,j)$ $[-]$ of the wetting phase in each block at time $t$, the pressure $P_t(i,j)$ [Pa] of the wetting phase in each block at time $t$, and the fluxes $q_t[(i_1,j_1) \rightarrow (i_2,j_2)]$ [ms$^{-1}$] of the wetting phase between neighboring blocks $(i_1,j_1)$ and $(i_2,j_2)$ at time $t$.

At each instant, the saturation in each block is updated according to the discretized mass balance law in the following way:

$$\frac{\theta}{\Delta t}\left[S_{t+\Delta t}(i,j) - S_t(i,j)\right] = \tag{1}$$
$$= \frac{1}{\Delta x}\left(q_t[(i-1,j) \rightarrow (i,j)] - q_t[(i,j) \rightarrow (i+1,j)] + q_t[(i,j-1) \rightarrow (i,j)] - q_t[(i,j) \rightarrow (i,j+1)]\right),$$

where $\theta$ $[-]$ denotes the porosity of the material, $\Delta t$ is a time step, and $\Delta x$ is the block size. A backward scheme can be also used (Kmec et al., 2021) but it slows the numerical algorithm unnecessarily.

The next step is to update the capillary pressure in each block according to the capillary pressure operator $P(S)$. The capillary pressure operator consists of the main wetting and draining branches defined by van Genuchten equation (5). To complete the capillary pressure operator, a hysteresis model is included (Parker and Lenhard, 1987). We use a similar approach to the play-type hysteresis used, e.g., in Rätz and Schweizer (2013); Schweizer (2017). All scanning curves are straight lines with a very large gradient $K_{PS}$. Once a block (in the wetting mode between the two main branches) reaches the main wetting branch along a scanning curve, it clings to it and continues along it. A similar procedure applies for the block in the draining mode.

Finally, the flux between neighboring blocks is updated according to the Darcy-Buckingham law (Bear, 1972):

$$q = \frac{\kappa}{\mu}k(S)\left(\rho g - \nabla P(S)\right), \tag{2}$$

where $\kappa$ [m$^2$] denotes the intrinsic permeability, $\rho$ [kgm$^{-3}$] the fluid density, $g$ [ms$^{-2}$] acceleration due to gravity, and $\mu$ [Pas] the dynamic viscosity of fluid, and $P(S)$ is the capillary pressure operator. The relative permeability function $k(S)$ is modelled by the form derived in Mualem (1976); Mualem and Dagan (1978); Van Genuchten (1980):

$$k(S) = S^\lambda\left[1 - \left(1 - S^{\frac{1}{m}}\right)^m\right]^2, \tag{3}$$

where $\lambda$ $[-]$ is a free parameter. Let us denote by $\gamma(S) = \kappa k(S)$ the effective permeability of the porous medium.

We use the following discrete implementation of the Darcy-Buckingham law (2):

$$q_t[(i_1,j_1) \rightarrow (i_2,j_2)] = \begin{cases} \frac{1}{\mu}\sqrt{\gamma(S_t(i_1,j_1)\gamma(S_t(i_2,j_2))}\left(\rho g - \frac{P_t(i_2,j_2) - P_t(i_1,j_1)}{\Delta x}\right) & \text{for } j_1 = j_2, \ i_2 = i_1 + 1 \\ \frac{1}{\mu}\sqrt{\gamma(S_t(i_1,j_1)\gamma(S_t(i_2,j_2))}\left(0 - \frac{P_t(i_2,j_2) - P_t(i_1,j_1)}{\Delta x}\right) & \text{for } i_1 = i_2, \ j_2 = j_1 + 1 \\ 0 & \text{otherwise} \end{cases} \tag{4}$$

Thus, for the hydraulic conductivity between blocks, we use the geometric mean of the conductivity values in the respective blocks. This choice of averaging has the desirable property of being small if the permeability of one of the blocks is small. The force of gravity is included only for the vertical fluxes $j_1 = j_2$. After setting the fluxes between neighboring blocks, the time is updated to $t + \Delta t$ and we proceed back to Eq. (1).

## 2.2 Scaling of the retention curve

A crucial idea behind the semi-continuum model is the appropriate scaling of the main branches of the retention curve which was first introduced by Vodák et al. (2022). The scaling of the retention curve is based on the fact that the shape of the retention curve depends on the size (especially the height) of the sample on which the measurement is made (Larson and Morrow, 1981; Hunt et al., 2013; Silva et al., 2018).

The simple scaling mechanism introduced in Vodák et al. (2022) is used here in which the main branches of the retention curve take the form of the standard van Genuchten model (Van Genuchten, 1980). More details about the proposed scaling of the retention curve and its sample size dependency are given in the following section. However, the detailed mathematical and physical justification is already published in Vodák et al. (2022), hence for a deeper understanding we refer to this paper.

For the reference block size $\Delta x_0$, the retention curve is modelled by the formula

$$p_0^w(S) = -\frac{1}{\alpha_w}\left(S^{\frac{-1}{m_w}} - 1\right)^{\frac{1}{n_w}}, \qquad\qquad p_0^d(S) = -\frac{1}{\alpha_d}\left(S^{\frac{-1}{m_d}} - 1\right)^{\frac{1}{n_d}}, \qquad\qquad (5)$$

where $S$ denotes saturation, $p_0^w$ is the capillary pressure on the wetting branch, $p_0^d$ is the capillary pressure on the draining branch, $\alpha_w, n_w$, and $m_w = 1 - \frac{1}{n_w}$ are parameters of the main wetting branch, and $\alpha_d, n_d$, and $m_d = 1 - \frac{1}{n_d}$ are parameters of the main draining branch.

The idea of the retention curve scaling is the following. For block size $\Delta x < \Delta x_0$, the retention curve becomes more flat but the distance between the main wetting and draining branches remains the same. Thus for the main wetting branch

$$p^w(S, \Delta x) = \frac{\Delta x}{\Delta x_0}p_0^w(S) + c^w \qquad\qquad (6)$$

with $c^w$ such that $p^w(0.5, \Delta x) = p_0^w(0.5)$, i.e.

$$c^w(\Delta x) = p_0^w(0.5)\left(1 - \frac{\Delta x}{\Delta x_0}\right). \qquad\qquad (7)$$

Clearly, for $\Delta x = \Delta x_0$, relation (6) reduces to equation (5) while for $\Delta x \to 0$, we obtain $p^w(S, \Delta x) \to p_0^w(0.5)$. For the main draining branch $p^d(S, \Delta x)$ the scaling is analogous to Eq. (6) and (7).

The scaling of the retention curve for 20/30 sand is shown in Fig. 1 for the reference block size $\Delta x_0 = \frac{10}{12}$ cm $\approx 0.83$ cm. Determining the dimension of $\Delta x_0$ is not trivial. It is explained in Sect. 3 how this dimension can be determined using the results of Bauters et al. (2000).

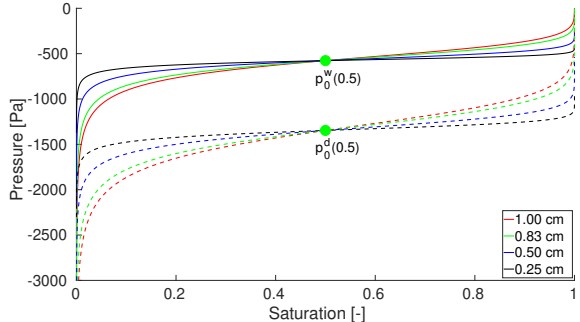

**Figure 1.** The scaling of the retention curve for $20/30$ sand. The solid lines denote the main wetting branches and the dashed lines denote the main draining branches for the respective value of $\Delta x$. The green curve represents the reference sample for $\Delta x_0 = \frac{10}{12}$ cm $\approx 0.83$ cm. As $\Delta x \to 0$, the retention curve becomes more flat but the distance between the main wetting and draining branches remains the same. The main wetting and draining branches "rotate" around fixed values $p_0^w(0.5)$ and $p_0^d(0.5)$, respectively.

## 2.3 Concept of the semi-continuum model and its limit in spatial variable

The scaling of the retention curve, i.e. the dependence of the capillary pressure-saturation relation on the block size, is not a common approach in flow modelling. However, the dependence of the experimentally determined retention curve on the porous medium sample size has been observed for a long time (Larson and Morrow, 1981; Mishra and Sharma, 1988; Zhou and Stenby, 1993; Perfect et al., 2004; Hunt et al., 2013; Ghanbarian et al., 2015). Note that this dependence on the sample volume also applies to other hydraulic and physical properties such as porosity or permeability (Mishra and Sharma, 1988; Ewing et al., 2010; Ghanbarian et al., 2017, 2021). In classical continuum mechanics, this scaling problem is "defined away" by the concept of the Representative Elementary Volume (REV). REV is the smallest volume for which the key physical quantities (e.g. saturation and pressure) can still be considered smooth. However, if the sample of porous medium is smaller than REV, key physical quantities, such as the retention curve, are strongly dependent on the sample size, and the continuum assumption is violated (Kouznetsova et al., 2001; White et al., 2006; Al-Raoush and Papadopoulos, 2010; Al-Raoush, 2012). The crucial idea of the semi-continuum model is to include the pressure-saturation dependency in the model, i.e. to scale the retention curve according to the block size. In the semi-continuum model, a block represents a real sample of the porous material. This makes the semi-continuum model fundamentally different from numerical schemes for solving partial differential equations where the block plays only a discretization (i.e. mathematical) role and regardless of the block size, the retention curve remains the same. In the semi-continuum model, the computational mesh (the blocks) takes into account the dependence of the physical parameters on the size of the blocks. Surprisingly, the idea of taking REV size into account in modelling porous media has been around for a long time. For instance, in White et al. (2006), the authors estimated the size of the REV and used it as a lower limit for the size of the finite elements. They argue that the use of smaller elements would lead to violation of continuum assumptions and thus the continuum approximation would no longer be appropriate. The same idea is used in the semi-continuum model: For blocks smaller than the REV, scaling of the retention curve must be included because the

continuum approximation is no longer adequate. Because we are interested in the description of flow phenomena below the REV scale, we need to include the dependence of the retention curve on the block size. This scaling of the retention curve must meet a physically justified requirement that the nature of the flow is preserved across all levels of block size. This means that the fluxes between neighboring blocks must not change when $\Delta x$ changes. Given Eq. (4), if $\Delta x$ decreases by half, the fluxes increase by a factor of two if the scaling of the retention curve is not included. Therefore, a linear scaling of the retention curve is introduced in Eq. (6), so the fluxes between blocks remain the same as $\Delta x$ decreases. For more details, see figures Fig. 4–6 in Vodák et al. (2022) that show the numerical convergence of the semi-continuum model in 1D and 2D.

The natural question is what the limit of the semi-continuum model would be as $\Delta x \to 0$. We tried to answer this question in Vodák et al. (2022) and derived the limit equation in a single spatial dimension:

$$(K_{PS}\partial_t S - \partial_t P_H)(P_H - v) \geq 0, \qquad \text{for all } v \in [C_2, C_1], \qquad \text{and } P_H \in [C_2, C_1],$$

$$\theta \partial_t S + \partial_x \left( \frac{\kappa}{\mu} \sqrt{k(S^-)}\sqrt{k(S^+)}(\rho g - \partial_x P_H) \right) = 0, \qquad S^{\pm}(x_0, t) = \lim_{x \to x_0^{\pm}} S(x, t). \tag{8}$$

In this equation, $\kappa$ denotes the intrinsic permeability, $\rho$ the fluid density, $g$ acceleration due to gravity, $\mu$ the dynamic viscosity of fluid, and $S$ the saturation. The values $C_1$ [Pa] and $C_2$ [Pa] denote the constant limits of the main wetting and draining branches, respectively. The limit is a partial differential equation containing a Prandtl-type hysteresis operator $P_H$ under the space derivative. If we are located on the main wetting or draining branches, the limit equation becomes a hyperbolic differential equation. Between the two main branches (i.e., we are located on the scanning curve), the limit represents a parabolic differential equation. It means the limit switches between parabolic and hyperbolic types of equation. The limit equation is a new type of mathematical model – we are not aware of any research that has investigated equations of this type. Note that the RE is a parabolic type equation – that is why it is only able to simulate the diffusion-like flow regime (Fürst et al., 2009).

## 3 Results

We want to completely reproduce the experiments reported in Bauters et al. (2000). The authors report that water was injected at a rate of $2\ \mathrm{cm}^3\ \mathrm{min}^{-1}$ through a hypodermic needle located near the sand surface. Thus, a point source infiltration is modeled such that a constant flux is prescribed across one centimeter of the top edge (in the middle). Zero discharge at the bottom boundary is prescribed, for simplicity. This choice of the bottom boundary condition does not affect the studied phenomena. All parameters used for the simulations are given in Table 1. The parameter $\lambda = 0.8$, which is consistent with experimental measurements (Schaap and Leij, 2000).

### 3.1 Adjustment of reference block size for 20/30 sand

First of all, the reference block size $\Delta x_0$ is unknown. This is a parameter of the semi-continuum model that has to be set. The parameter $\Delta x_0$ was calibrated by simulating the experiments of Bauters et al. (2000). In the simulation, we use the parameters for 20/30 sand adopted from Schroth et al. (1996) and DiCarlo (2004) (see Table 1). We ran several simulations of the semi-continuum model with $\Delta x_0$ equal to $\frac{10}{12}$ cm, 1.00 cm and $\frac{12}{10}$ cm. The moisture profile was calculated for three different initial

| Parameter | Symbol | Value |
|---|---|---|
| Horizontal width of the chamber | $A$ | 31 cm |
| Vertical length of the chamber | $B$ | 50 cm |
| Reference block size | $\Delta x_0$ | 0.83 cm |
| Block size | $\Delta x$ | 0.25 cm |
| Porosity | $\theta$ | 0.35 |
| Density of water | $\rho$ | $1000 \text{ kgm}^{-3}$ |
| Dynamic viscosity of water | $\mu$ | $9 \times 10^{-4} \text{ Pas}$ |
| Intrinsic permeability | $\kappa$ | $2.294 \times 10^{-10} \text{ m}^2$ |
| Relative permeability exponent | $\lambda$ | 0.8 |
| Acceleration due to gravity | $g$ | $9.81 \text{ ms}^{-2}$ |
| Wetting curve parameter | $\alpha_w$ | $0.177 \text{ cm}^{-1}$ |
| Wetting curve parameter | $n_w$ | 6.23 |
| Draining curve parameter | $\alpha_d$ | $0.0744 \text{ cm}^{-1}$ |
| Draining curve parameter | $n_d$ | 8.47 |
| Slope of scanning curves | $K_{PS}$ | $10^5 \text{ Pa}$ |
| Boundary flux | $q_B$ | $8 \times 10^{-5} \text{ ms}^{-1}$ |

**Table 1.** Parameters used to reproduce the experiments of Bauters et al. (2000). Parameters for 20/30 sand were adopted from Schroth et al. (1996) and DiCarlo (2004).

saturation: a dry (0.001), a medium dry (0.01) and a wet (0.05) porous medium. The parameters used for simulations are given in Table 1, except for the block size $\Delta x = 0.50$ cm and $A = 17$ cm (horizontal width of the chamber), which were changed in order for the simulations not to be extremely time consuming. The moisture profiles for all values of $\Delta x_0$ are depicted in Fig. 2.

We want to choose the parameter $\Delta x_0$ for which the non-monotonic behavior of the moisture profiles widths occurs. Table 2 shows the width of the moisture profiles. The width of the moisture profile is calculated in the following way: First, we calculate the width of each row, which equals $n_{row} \times \Delta x$, where $n_{row}$ is a number of blocks in the row for which the saturation exceeds 0.07 during the simulation and $\Delta x$ is the size of the block. The width of the moisture profile is then calculated as the average width of all rows with non-zero width. It is clear that the most pronounced non-monotonic behavior of the moisture profiles widths is given for $\Delta x_0 = \frac{10}{12} \approx 0.83$ cm (Fig. 2a), and is therefore the most appropriate.

Note that the width of the finger is not constant for initially dry porous medium, although it is experimentally observed (Bauters et al., 2000). This artificial behavior is due to the unrealistic homogeneity of porous medium used for the simulation. Although, in reality, the porous medium is homogeneous, this does not mean that all the characteristics are identical in each block of the simulation. If a small distribution of the intrinsic permeability is included, the finger width will be constant. This is demonstrated in the next section.

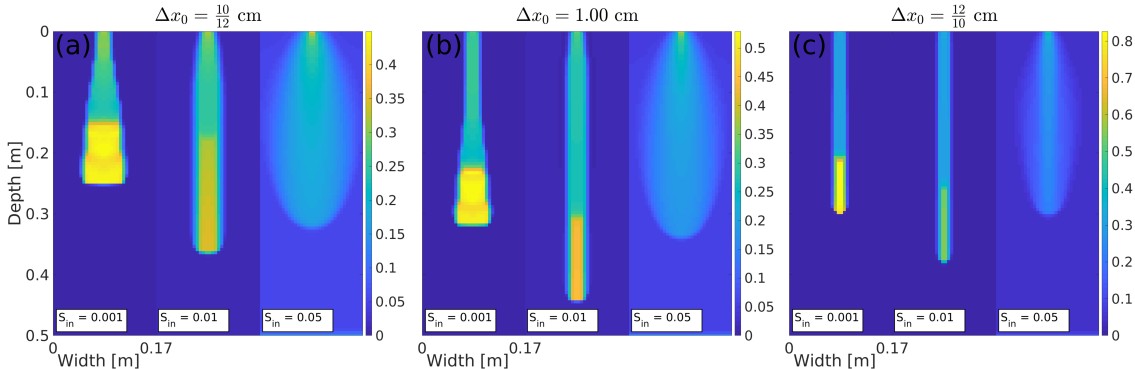

**Figure 2.** Snapshots of the saturation field for various $\Delta x_0$ for initially dry ($S_{in} = 0.001$), a medium dry ($S_{in} = 0.01$) and a wet ($S_{in} = 0.05$) porous material. The moisture profiles for **(a)** $\Delta x_0 = \frac{10}{12}$ cm, **(b)** $\Delta x_0 = 1.00$ cm and **(c)** $\Delta x_0 = \frac{12}{10}$ cm are shown at 30, 30 and 20 minutes, respectively. Saturation values are colour-coded according to the colour bar on the right.

| reference block size $\Delta x_0$ | moisture profile width for: | | |
| --- | --- | --- | --- |
| | $S_{in} = 0.001$ | $S_{in} = 0.01$ | $S_{in} = 0.05$ |
| $\frac{10}{12}$ cm | 5.3137 cm | 4.6986 cm | 8.1000 cm |
| 1.00 cm | 4.0156 cm | 3.8202 cm | 7.8500 cm |
| $\frac{12}{10}$ cm | 2.6333 cm | 2.8421 cm | 4.6200 cm |

**Table 2.** The width of the moisture profiles for different values of $\Delta x_0$.

## 3.2 Wetting front dependence on initial saturation

Let us now demonstrate the ability of the semi-continuum model to capture the Bauters' paradox. As mentioned above, even in homogeneous porous medium, all characteristics are not identical in each block. Thus, the spatially correlated distribution of the intrinsic permeability is introduced. Such distribution was also used e.g in Kmec et al. (2021). The distribution satisfies $\kappa_{max}/\kappa_{min} \approx 4$ and the mean of the intrinsic permeability approximately equals $\kappa$. The distribution of the values of intrinsic permeability is shown in Fig. 3. The distribution of the intrinsic permeability is not the cause of the Bauters' paradox. However, with such a distribution, more physical-looking fingers evolve. For a simulation of the Bauters' paradox without the intrinsic permeability distribution, see Fig. A1 in Appendix A.

Figure 4 shows a snapshot of the saturation field at 25 minutes for seven different values of the initial saturation. It can be seen that as the initial saturation increases, the finger first gets faster and narrower. Then the finger slows down and widens and finally gives way to a diffusion-like plume. The transition between unstable and stable flow is also in agreement with the experimental observation: The non-monotonic behavior of the finger width and velocity is captured correctly as well as the shape of the wetting front. Moreover, a stable wetting front appears for initial saturation higher than 0.03, which is also

255 consistent with experiments. Note that the authors of Bauters et al. (2000) only recorded the wetting front patterns 15 cm from the top. Therefore, we are not able to compare the wetting fronts at the upper part of the chamber.

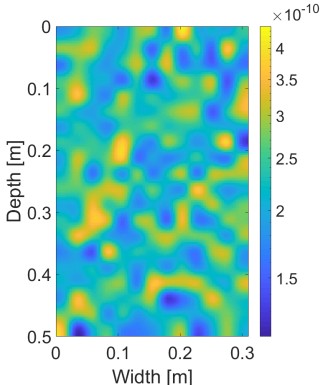

**Figure 3.** The distribution of the intrinsic permeability $\kappa$ [m$^2$], which satisfies $\kappa_{max}/\kappa_{min} \approx 4$. Intrinsic permeability values are colour-coded according to the colour bar on the right.

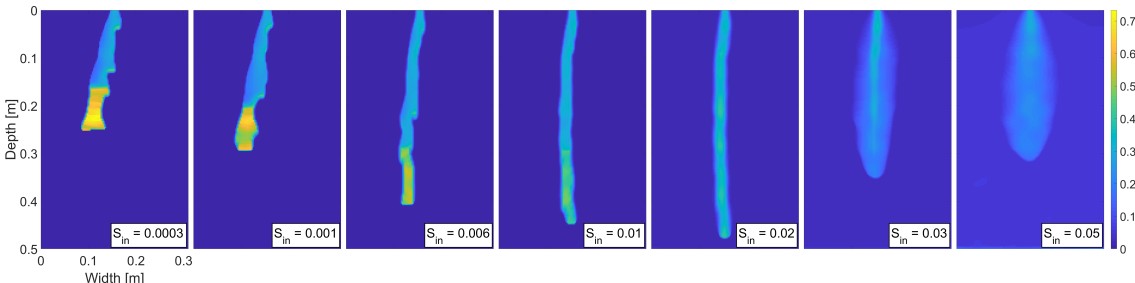

**Figure 4.** Snapshot of the saturation field at 25 minutes for seven different values of the initial saturation. Saturation values are colour-coded according to the colour bar on the right. Initial saturation of the medium increases from left to right.

One may wonder if this complex behavior depends on the choice of the intrinsic permeability distribution. We generated seven different distributions (see Fig. A2) and the same simulations as above were performed. Snapshots of the saturation field 25 minutes from the beginning of the infiltration for seven different values of the initial saturation are shown in Fig. A3 - A9.

260 The figures show that the character of the flow remains the same for all types of distributions. Thus, the distribution of the intrinsic permeability does not affect the transition from the finger flow to the diffusion-like flow.

### 3.3 Width and velocity of the fingers

Figure 5a shows the width of the fingers (moisture profiles) 25 minutes from the beginning of infiltration for the simulation shown in Fig. 4 (wetting profiles for $S_{in} = 0.0005$, 0.002, 0.04 are not included in Fig. 4 to make the figure more readable).

265 The width of each moisture profile is calculated in the same way as was used in Table 2. We can clearly see that the finger

width first slightly decreases and then increases. The narrowest finger is produced for $S_{in} = 0.01$ (2.70 cm) which is consistent with experiments (see Fig. 5 in Bauters et al. (2000)). Let us note that the finger width for $S_{in} = 0.0003$ (3.74 cm) is slightly smaller than for $S_{in} = 0.0005$ (3.82 cm). However, this is due to the distribution of the intrinsic permeability. Indeed, the finger width for all simulations given by eight different distributions of the intrinsic permeability (see Fig. 3 and Fig. A2) is depicted in Fig. 5b. We observe that – on average – the finger width for the lowest initial saturation used in the simulation is higher than for $S_{in} = 0.0005$.

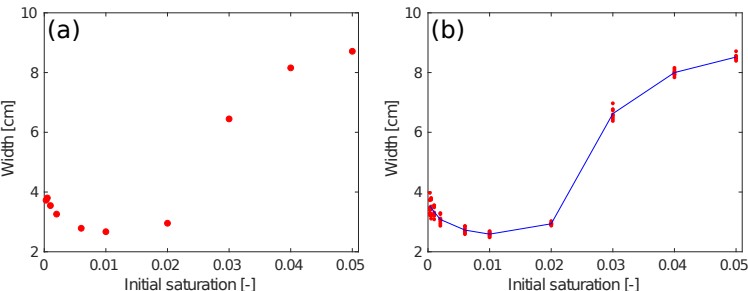

**Figure 5.** The width of the finger (or wetting front) at $t = 25$ minutes is plotted against the initial saturation. **(a)** For the distribution of the intrinsic permeability given by Fig. 3. **(b)** For all simulations given by eight different distributions of the intrinsic permeability (see Fig. 3 and Fig. A2). The blue line connects the averages.

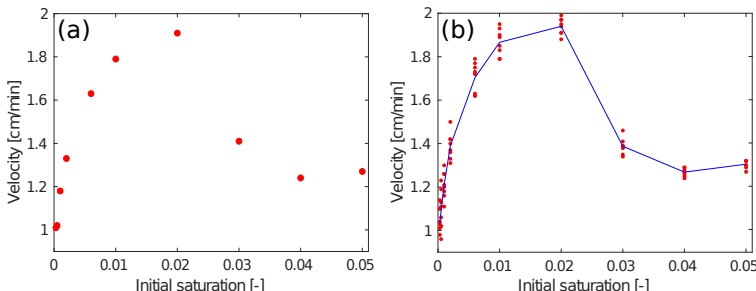

**Figure 6.** The velocity of the wetting front at $t = 25$ minutes is plotted against the initial saturation. **(a)** For the distribution of the intrinsic permeability given by Fig. 3. **(b)** For all simulations given by eight different distributions of the intrinsic permeability (see Fig. 3 and Fig. A2). The blue line connects the averages.

For finger velocity, we proceed similarly, i.e., we find the bottom-most block of the finger whose saturation exceeds 0.07. The depth of the bottom-most block defines the current length of the finger. Finger velocity is computed as the rate of change of the finger length in time. The finger (or wetting front) velocity at $t = 25$ minutes for the simulation given by Fig. 4 is summarized in Fig. 6a. The advance of the wetting front was slower for the diffusion-like behavior compared to finger flow (but higher than for $S_{in} = 0.002$). This is rather counter-intuitive, since the classical theory as the Richards' Equation predicts an increase in velocity with increasing initial saturation. The highest finger velocity is observed for $S_{in} = 0.02$, and it is approximately five

times lower than the highest finger velocity experimentally observed in Bauters et al. (2000) (for $S_{in} = 0.01$). This is consistent because we used four times lower influx in our simulations compared to the experiments. We observed that the character of the dependence remains the same for different distributions of the intrinsic permeability (see Fig. 6b).

## 3.4 Water content at and behind the wetting front

Let us now examine the change in saturation at and behind the wetting front (a finger tip). The difference between the saturation of the tip and the tail is called the saturation overshoot magnitude. To quantify the saturation overshoot magnitude, the saturation is averaged for each row, which gives the saturation profiles in 1D. Averaging is applied only to those blocks whose saturation exceeds 0.07. Saturation overshoot magnitude is then given as an average saturation at the finger tip minus an average saturation at the finger tail. In the case of diffusion-like flow with no overshoot, we average the bottom 20 centimeters of the saturation profile and subtract the average of the rest of the profile.

The dependence of saturation overshoot magnitude on initial saturation at $t = 25$ minutes is shown in Fig. 7. We see that there is a hyperbolic decay relationship between the initial saturation and the saturation overshoot magnitude ($R^2 = 0.990$). This is consistent with the experimental observation (Bauters et al., 2000). There is still a minor saturation overshoot for $S_{in} = 0.02$. This is again consistent with the experiments, where the authors observed a saturation overshoot for $S_{in} = 0.02$, but no overshoot for $S_{in} = 0.03$.

Let us note that the distribution of the intrinsic permeability causes higher variability in the saturation profiles. Without this distribution, the accuracy of the fit is better. This was shown in the 1D semi-continuum model, where the hyperbolic fit was obtained with $R^2 = 0.995$ (see Fig. 3.6. in Kmec (2021)).

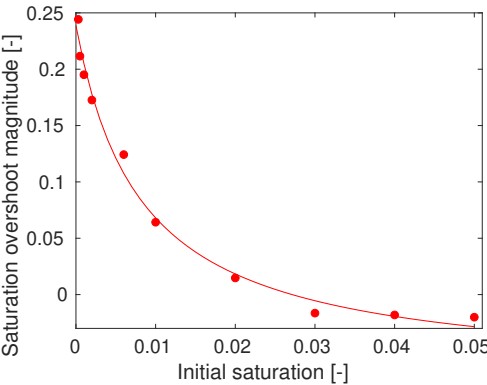

**Figure 7.** Dependence of the saturation overshoot magnitude on initial saturation at 25 minutes. Saturation overshoot magnitude is computed even for diffusion-like profiles (see the text for the methodology). A hyperbolic relation $f(x) = \frac{ax+b}{cx+d}$ fitted to the simulated data has a $R^2$ value of 0.990.

## 3.5 Sensitivity analysis

The parameters such as infiltration rate and material characteristics are not fitted to obtain the best results. Let us first demonstrate the effect of the boundary flux on the flow regime. Since all the simulations are computationally demanding, a larger block size $\Delta x = 0.50$ cm is used while the rest of parameters remained the same (see Table 1).

Five different values of the boundary flux $q_B$ were used ranging from $2 \times 10^{-5}$ ms$^{-1}$ to $16 \times 10^{-5}$ ms$^{-1}$. The baseline simulations are given for $q_B = 8 \times 10^{-5}$ ms$^{-1}$. For each value of $q_B$, 28 different simulations are performed, with variable initial saturation (seven different initial saturation) and variable intrinsic permeability distribution (four different distributions; see Fig. B1). Note that 140 different simulations were performed in total. The same scheme was applied to all other sensitivity analysis simulations, i.e. the analysis was always performed for seven different values of initial saturation and four different

distributions of intrinsic permeability. Figure 8 shows the width and the velocity of the fingers (moisture profiles) for five different values of boundary flux $q_B$ and for seven different values of the initial saturation $S_{in}$. For a given value $q_B$ and $S_{in}$, the average width and velocity of the four different distributions of the intrinsic permeability is calculated and plotted.

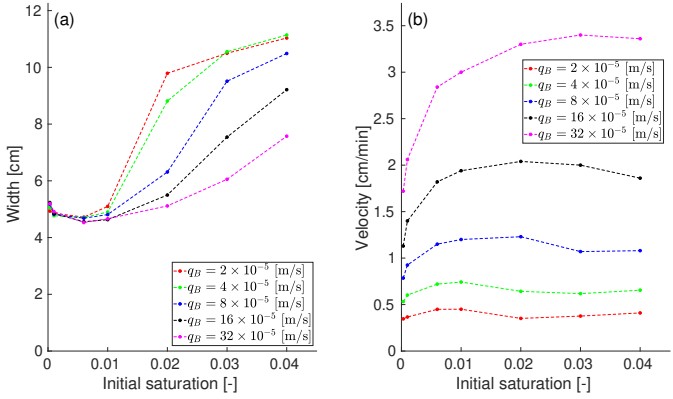

**Figure 8.** The effect of the boundary flux on the flow regime. The width **(a)** and the velocity **(b)** of the finger (or wetting front) is plotted against the initial saturation for five different values of boundary flux $q_B$. Times are scaled according to the boundary flux $q_B$, thus the width and velocity of the moisture profiles are calculated at $t = 100, 50, 25, 12.5$ and $6.25$ minutes for $q_B = 2, 4, 8, 16, 32 \times 10^{-5}$ ms$^{-1}$, respectively. The averages of four different distributions of the intrinsic permeability are plotted.

Since the used boundary fluxes varied by more than one order of magnitude, the times for which the velocity and the width are calculated need to be scaled according to the boundary flux. The time-points used are $t = 100, 50, 25, 12.5$ and $6.25$

minutes for $q_B = 2, 4, 8, 16, 32 \times 10^{-5}$ ms$^{-1}$, respectively. It can be seen in Fig. 8a that with decreasing boundary flux, the flow tends to become more diffusive. The transition between the finger-like and diffusion-like regimes is clearly evident for the initial saturation, for which the width of the moisture profiles increases rapidly. For instance, for $q_B = 2 \times 10^{-5}$ ms$^{-1}$ and $q_B = 4 \times 10^{-5}$ ms$^{-1}$, the rapid increase can be already seen for $S_{in} = 0.02$. For higher values of $q_B$, diffusion-like behavior is observed for higher values of initial saturation. To make this as clear as possible, a snapshot of the saturation field for the

intrinsic permeability distribution defined by Fig. B1a is shown in the left panel of Fig. B5. Note that the dependence on the boundary flux is in good agreement with the experimental observation (DiCarlo, 2004). Moreover, it is not surprising that the velocity of the moisture profiles shown in Fig. 8b decreases with decreasing boundary flux.

It is evident that the non-monotonic behavior of the width and the velocity of the moisture profiles is not dependent on the boundary flux. Hence, the Bauters' paradox is observed for all tested values of $q_B$. Since the diffusion-like behavior occurs

at lower values of boundary flux, the manifestation of the Bauters' paradox is shifted to higher initial saturation values as the boundary flux increases.

In order not to extend the main part of the manuscript too much, a sensitivity analysis for other parameters such as intrinsic and relative permeability, dynamic viscosity and retention curve is included in Appendix B. It is shown that the Bauters' paradox occurs for different values of material parameters. For details, see Fig. B1 – Fig. B6 and the corresponding text.

## 4  Discussion

To our best knowledge, the presented semi-continuum model is the first model which is able to fully capture the Bauters' paradox. This is achieved without introducing any new parameters, or material functions. The semi-continuum model is based on well established physics only – mass balance equation, the Darcy-Buckingham law, and a proper scaling of the retention curve with the volume of the block. The model may help to explain the precise mechanism of the transition between the

finger-like and diffusion-like regimes.

We conjecture that the explanation of the Bauters' paradox is rather similar to the non-monotonic dependence of porous medium flow on the magnitude of the influx. For very small values of influx, the flow becomes stable with increasing finger width. The same applies for very large values of influx. Hence, the unstable flow is only observed for fluxes within a certain range (Yao and Hendrickx, 1996; Glass et al., 1989b; DiCarlo, 2013). Yao and Hendrickx (1996) hypothesized that the stable

flow occurs when the effect of gravity becomes negligible. This happens in two "extreme" cases. First, at very low infiltration rates, capillarity becomes the dominant force compared to the force of gravity. Second, for infiltration rates higher than the saturated hydraulic conductivity, the viscosity dominates and the stable flow without fingers occurs. In our case, the dependence of the flow regime on the initial saturation behaves similarly. For initially dry porous medium, the capillarity dominates and the large capillary forces are able to win over gravity in sucking the water sideways into dry areas of the matrix. In a medium,

which is moderately wet, this becomes more difficult, because the capillary forces are generally lower. Thus, in a moderately wet medium, the fingers become thinner and faster. At sufficiently high initial saturation, the large conductivity between neighboring blocks prevents water piling up behind the wetting front and the formation of saturation overshoot. This results in the ability of lateral expansion because the persistence of the fingers is suppressed (Rezanezhad et al., 2006; Kmec et al., 2021). Therefore, a diffusion-like flow regime is observed.

A distribution of intrinsic permeability was used in the model. This was motivated by the following observation: As the blocks get smaller and smaller, the variability of their material characteristics necessarily increases. The characteristic of a block is given by an average over the pores of the block. As the block size decreases, so does the number of pores over which

the average is taken. Thus, the variability of the characteristics increases. It is possible to introduce a distribution of other parameters such as porosity and the parameters of the retention curve (White et al., 2006; Ghanbarian et al., 2021). However, to keep the model as simple as possible, this has not been implemented here. It should be stressed that the Bauters' paradox appears even if the intrinsic permeability is kept homogeneous. Furthermore, a sensitivity analysis of the Bauters' paradox was performed, which showed that the Bauters' paradox occurs for different values of material parameters and boundary flux.

DiCarlo states the following four criteria to evaluate a model for unsaturated porous media flow (DiCarlo, 2013). Paraphrasing his words, the model should

1. have a minimum of adjustable parameters, and the parameters should be meaningful,

2. reduce to the RE in non-overshoot and static profiles,

3. produce a good match of the observed 1D profiles, not just the magnitude of the overshoot,

4. be able to produce predictions of the 2D and 3D preferential flow in terms of finger widths and finger spacing.

Since the RE can simulate only a diffusion-like regime, we understand (2) in the way that the model should be able to reproduce also diffusion-like regime, not only the fingering regime. This does not mean that the semi-continuum model behaves in the same way as the RE in non-overshoot profiles. This is of course not possible due to the scaling of the retention curve.

The semi-continuum model formulation uses only the physics of the Richards' Equation (porosity, permeability, the pressure-saturation relation, mass conservation, and the Darcy-Buckingham Law). The block size used in the simulation is not a free parameter – it is tied to the retention curve by the scaling relation and the reference block size $\Delta x_0$. The value of $\Delta x_0$ is not arbitrary, it is connected to the REV. Thus, item 1 of DiCarlo's list is satisfied. In view of Fig. 4, 5, 6, 7 and the results in Kmec et al. (2019, 2021), we claim items 2–4 are also satisfied.

However, there are two exceptions: For 2D preferential flow, the dependence of finger width and finger spacing on the influx is still missing. Here, we mention, for example, the experiments of Yao and Hendrickx (1996) for low infiltration rates and Glass et al. (1989b) for higher infiltration rates. We will discuss this complex dependency in a forthcoming paper. Moreover, the semi-continuum model has not yet been extended to 3D.

Note that for a given $\Delta x$ and retention curve, the semi-continuum model may look like a numerical scheme for the RE. However, when using different block size $\Delta x$, a different retention curve must be used for the RE to retain the character of the flow. Otherwise, only the diffusion-like behavior occurs (Fürst et al., 2009). In contrast, for the semi-continuum model, we define the retention curve for the reference block size $\Delta x_0$ and the retention curve is then scaled automatically according to the block size. In this case, the retention curve is a measurable material characteristic. The semi-continuum model is thus predictive; we do not need to fit the retention curve for each $\Delta x$ separately. Therefore, the semi-continuum model is not a numerical scheme to solve RE. The crucial difference between the semi-continuum model and a numerical scheme for the RE is in an appropriate scaling of the retention curve with the block size. As $\Delta x$ decreases, the semi-continuum model retains the character of the flow between the blocks and the saturation overshoot does not disappear (Vodák et al., 2022). Let us also stress that it is important to use the geometric mean of the hydraulic conductivity for computing the flux between neighboring blocks. In principle, it is necessary to use a type of averaging that has the desirable property of being small if the permeability

of one of the blocks is small. Such an averaging of the hydraulic conductivity creates a pile-up effect, resulting in a finger with saturation overshoot. Thus, the geometric mean is not only possible averaging choice; for example, the harmonic mean can also be used. In the semi-continuum model, we use the geometric mean because it is shown that using this type of averaging is

the most appropriate in the case of random pore networks (Jang et al., 2011).

We can summarize the role of (1) the appropriate averaging the hydraulic conductivity (for instance the geometric mean) and (2) the scaling of the retention curve as follows: The geometric mean is essential to create the pile-up effect (saturation overshoot), while the effect of scaling the retention curve is to preserve this saturation overshoot for $\Delta x \to 0$. If (1) and (2) were not utilized in the semi-continuum model, diffusion-like flow patterns would always be produced with a monotonic saturation

profile. This behavior is demonstrated for the initially almost dry medium in Fig. 9. The used distribution of the intrinsic permeability is shown in Fig. 3. A typical finger with saturation overshoot is produced for the semi-continuum model (Fig. 9a), while without (1) and (2), a monotonic diffusion-like profile is formed (Fig. 9b). In Vodák et al. (2022) we have demonstrated that the overshoot behavior is not lost in the limit $dx \to 0$; for the numerical convergence see figures Fig 4 − 6 in Vodák et al. (2022). Thus, the semi-continuum model does not converge to the RE, even if the block size goes to zero. It converges to a

new type of hysteretic partial differential equation defined by Eq. (8) that – to our knowledge – has not been studied so far. We invite the porous media community to study the semi-continuum model and its limit because so far, it has been proven to capture well all of the complex and counter-intuitive features of unsaturated homogeneous porous media flow that have been observed and reported in the literature.

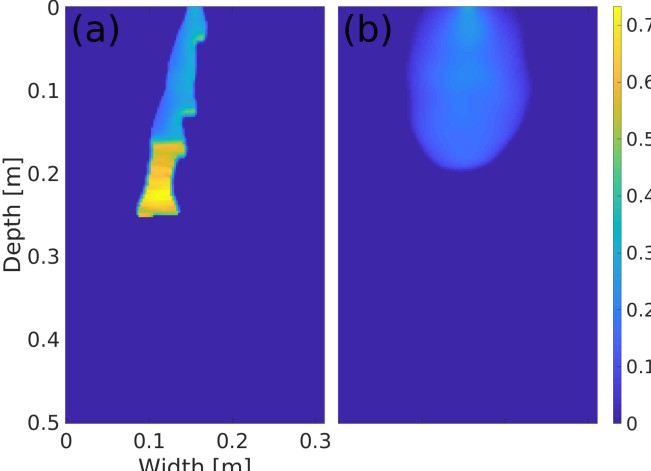

**Figure 9.** The role of the geometric mean of the hydraulic conductivity and the scaling of the retention curve. Snapshot of the saturation field for **(a)** the semi-continuum model and **(b)** without utilizing the geometric mean of the hydraulic conductivity and the scaling of the retention curve at $t = 25$ minutes for $S_{in} = 0.0003$. Saturation values are colour-coded according to the colour bar on the right.

## 5    Conclusions

It is known from infiltration experiments that unsaturated porous media flow patterns depend on the initial saturation of the medium in a complex way. Going from initially dry to initially wet medium, the flow pattern changes from finger-like regime with a pronounced saturation overshoot to a diffusion-like regime with no overshoot. During the transition, several finger characteristics (velocity, overshoot magnitude, finger width) change in a non-monotonic way. This complex behavior is called the Bauters' paradox and the standard continuum mechanics-based theory has been unable to reproduce it.

Here, we introduced a semi-continuum model (discrete in space, and continuous in time) which is able to correctly reproduce all the observed features of the Bauters' paradox. The semi-continuum model implements a physically relevant scaling of the retention curve – the slope of the retention curve decreases with decreasing block size. This model correctly reproduces the flow patterns both for initially dry, and initially wet porous medium.

### Appendix A

Figure A1 shows a snapshot of the saturation field at 25 minutes for seven different values of the initial saturation. The distribution of the intrinsic permeability is not included, i.e. the medium is perfectly homogeneous. The effect of the intrinsic permeability distribution is pronounced for the initially dry porous medium, while for the initially wet porous medium this effect is negligible. This is expected because in the case of diffusion-like regime, small changes in intrinsic permeability do not have a significant effect on the flow. The artificial looking behavior for the initially dry porous medium is eliminated if a more

realistic porous medium is used for the simulations, i.e. if the distribution of the intrinsic permeability is included.

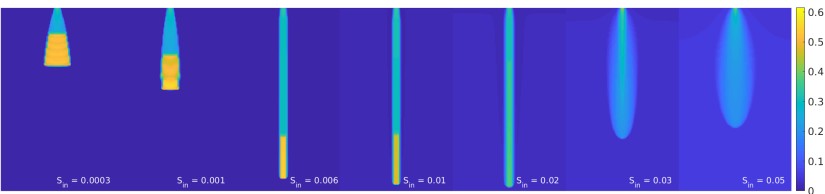

**Figure A1.** Snapshot of the saturation field at 25 minutes for seven different values of the initial saturation. The distribution of the intrinsic permeability is not included. Saturation values are colour-coded according to the colour bar on the right. Initial saturation of the medium increases from left to right.

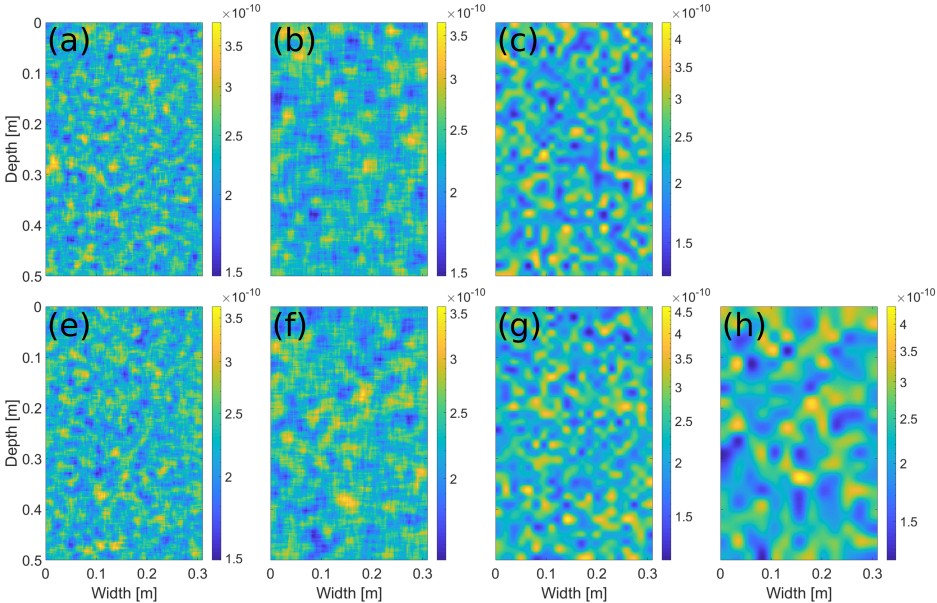

**Figure A2.** The distribution of the intrinsic permeability $\kappa$ [m$^2$]. The distributions satisfy: **(a)** $\kappa_{max}/\kappa_{min} \approx 2.60$, **(b)** $\kappa_{max}/\kappa_{min} \approx 2.50$, **(c)** $\kappa_{max}/\kappa_{min} \approx 3.40$, **(e)** $\kappa_{max}/\kappa_{min} \approx 2.45$, **(f)** $\kappa_{max}/\kappa_{min} \approx 2.35$, **(g)** $\kappa_{max}/\kappa_{min} \approx 3.90$, **(h)** $\kappa_{max}/\kappa_{min} \approx 3.50$. Intrinsic permeability values are colour-coded according to the colour bar on the right.

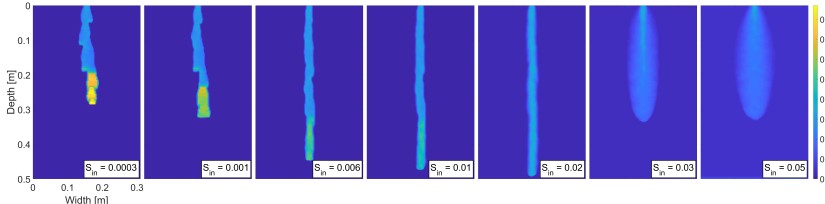

**Figure A3.** Snapshot of the saturation field at 25 minutes for seven different values of the initial saturation for the distribution which satisfies $\kappa_{max}/\kappa_{min} \approx 2.60$ (the distribution in Fig. A2a). Saturation values are colour-coded according to the colour bar on the right.

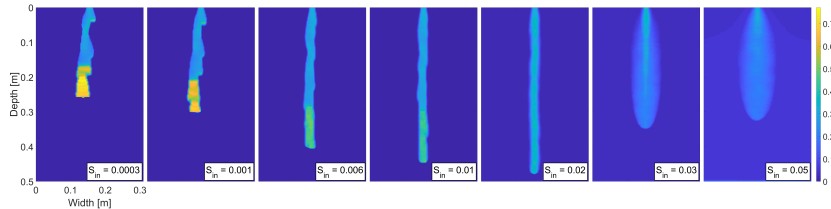

**Figure A4.** Snapshot of the saturation field at 25 minutes for seven different values of the initial saturation for the distribution which satisfies $\kappa_{max}/\kappa_{min} \approx 2.50$ (the distribution in Fig. A2b). Saturation values are colour-coded according to the colour bar on the right.

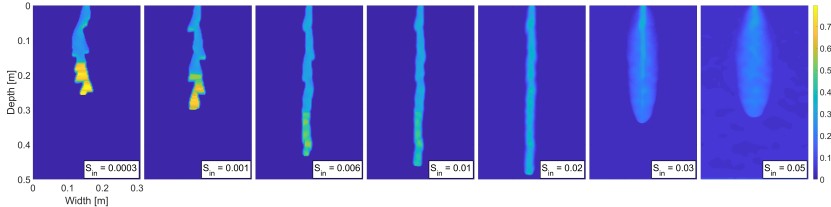

**Figure A5.** Snapshot of the saturation field at 25 minutes for seven different values of the initial saturation for the distribution which satisfies $\kappa_{max}/\kappa_{min} \approx 3.40$ (the distribution in Fig. A2c). Saturation values are colour-coded according to the colour bar on the right.

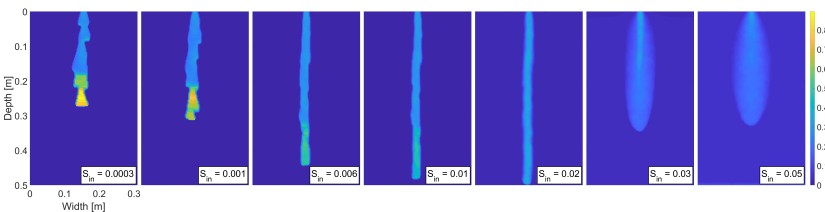

**Figure A6.** Snapshot of the saturation field at 25 minutes for seven different values of the initial saturation for the distribution which satisfies $\kappa_{max}/\kappa_{min} \approx 2.45$ (the distribution in Fig. A2e). Saturation values are colour-coded according to the colour bar on the right.

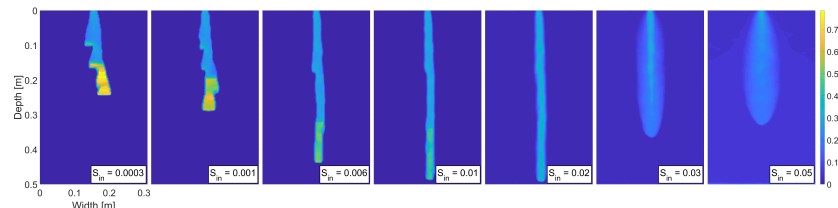

**Figure A7.** Snapshot of the saturation field at 25 minutes for seven different values of the initial saturation for the distribution which satisfies $\kappa_{max}/\kappa_{min} \approx 2.35$ (the distribution in Fig. A2f). Saturation values are colour-coded according to the colour bar on the right.

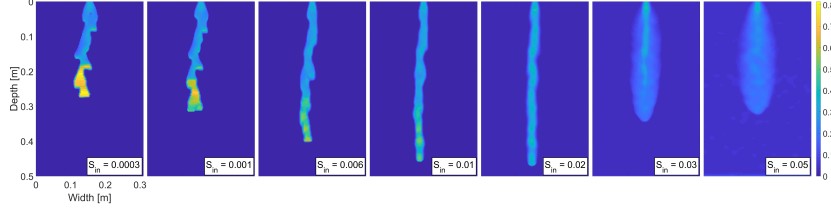

**Figure A8.** Snapshot of the saturation field at 25 minutes for seven different values of the initial saturation for the distribution which satisfies $\kappa_{max}/\kappa_{min} \approx 3.90$ (the distribution in Fig. A2g). Saturation values are colour-coded according to the colour bar on the right.

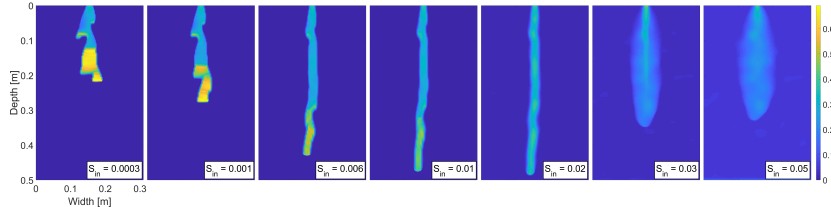

**Figure A9.** Snapshot of the saturation field at 25 minutes for seven different values of the initial saturation for the distribution which satisfies $\kappa_{max}/\kappa_{min} \approx 3.50$ (the distribution in Fig. A2h). Saturation values are colour-coded according to the colour bar on the right.

## Appendix B

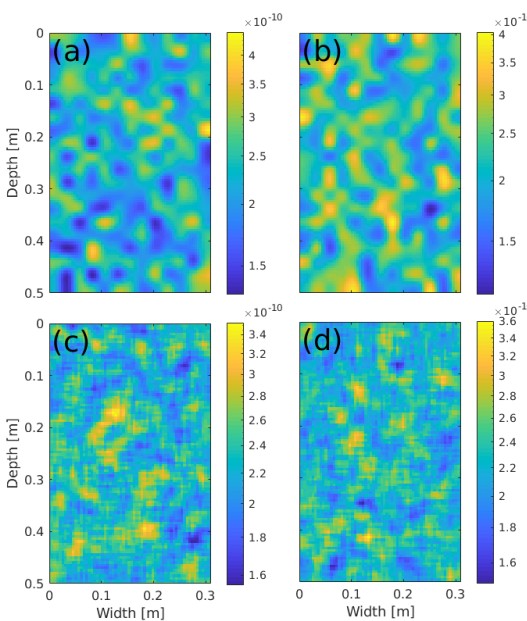

**Figure B1.** The distribution of the intrinsic permeability $\kappa$ [m$^2$] used for the sensitivity analysis of the Bauters' paradox. The distributions satisfy: **(a)** $\kappa_{max}/\kappa_{min} \approx 3.40$, **(b)** $\kappa_{max}/\kappa_{min} \approx 3.48$, **(c)** $\kappa_{max}/\kappa_{min} \approx 2.28$, **(d)** $\kappa_{max}/\kappa_{min} \approx 2.42$. Intrinsic permeability values are colour-coded according to the colour bar on the right.

### Effect of the intrinsic permeability and dynamic viscosity on the flow regime

Increasing the intrinsic permeability $\kappa$ has the same effect as decreasing the parameter $\mu$ and vice versa. Therefore, a fraction $\frac{\kappa}{\mu}$ is used for the sensitivity analysis of both these parameters. The baseline values of $\kappa$ and $\mu$ are given in Table 1. Five different values $b \cdot \frac{\kappa}{\mu}$ were examined, where $b = 0.50, 0.75, 1.00, 1.50$ and $2.00$. Obviously, baseline simulations are given for $b = 1.00$. Figure B2 shows the width and the velocity of the fingers (moisture profiles) 25 minutes from the beginning of infiltration for

five different values of $b$ and for seven different values of initial saturation $S_{in}$. For a given value $b$ and $S_{in}$, the average width and velocity of the moisture profile of four different distributions of the intrinsic permeability were calculated and plotted.

It can be seen in Fig. B2a that as parameter $b$ increases, the width of the moisture profiles increases for higher initial saturation. For lower initial saturation, the effect of $b$ is negligible. This is because with increasing parameter $b$, diffusion-like behavior is observed for lower values of initial saturation. The transition between the finger-like and diffusion-like regimes is clearly evident for the initial saturation, for which the width of the moisture profiles increases rapidly. For instance, for $b = 2.00$ and $b = 1.50$, the rapid increase can be already seen for $S_{in} = 0.02$. For lower values of $b$, diffusion-like behavior is observed for higher values of initial saturation. For clarity, a snapshot of the saturation field at 25 minutes for the intrinsic permeability distribution defined by Fig. B1a is shown in the right panel of Fig. B5. As for the moisture profile width, the velocity of the moisture profiles also increases with increasing $b$ as can be seen in Fig. B2b. This is expected because the parameter $b$ affects directly the magnitude of the flow.

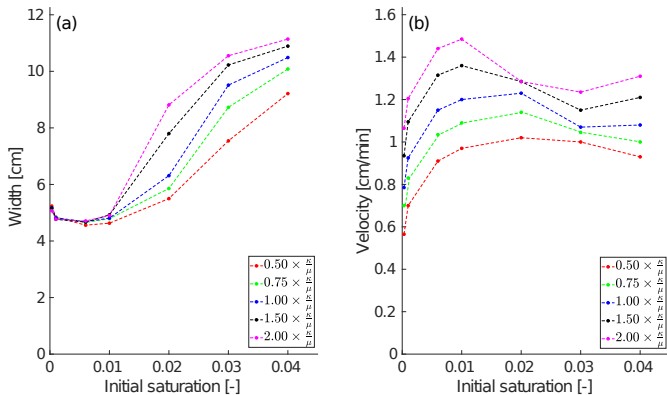

**Figure B2.** The effect of the intrinsic permeability and dynamic viscosity on the flow regime. The width **(a)** and the velocity **(b)** of the finger (or wetting front) at $t = 25$ minutes is plotted against the initial saturation for five different values of $b \cdot \frac{\kappa}{\mu}$. The average of four different distributions of the intrinsic permeability is plotted.

Finally, it can be seen that the Bauters' paradox is observed for all values of $b$. Therefore, the non-monotonic behavior of the width and the velocity of the moisture profiles is not dependent on intrinsic permeability and/or dynamic viscosity.

**Effect of relative permeability on the flow regime**

The relative permeability function $k(S)$ is given by Eq. (3). The function contains a free parameter $\lambda[-]$ and therefore the effect of the relative permeability on the flow regime is tested by using five different values of $\lambda$ ranging from 0.6 to 1.0. The baseline simulations are given for $\lambda = 0.8$. Note that the parameter $\lambda$ affects the value of the relative permeability especially for an initially dry porous medium. For the lowest initial saturation used for simulations ($S_{in} = 0.0003$), the relative permeability is more than 25 times larger for $\lambda = 0.6$ compared to $\lambda = 1.0$. In contrast, it is approximately 3.6 times larger for $S_{in} = 0.04$. As mentioned above, 28 different simulations were performed for each $\lambda$ with variable initial saturation and intrinsic permeability

distribution. Again, a snapshot of the saturation field at 25 minutes for the intrinsic permeability distribution defined by Fig. B1a is shown in the left panel of Fig. B6.

Figure B3 shows the width and the velocity of the fingers (moisture profiles) 25 minutes from the beginning of infiltration for five different values of $\lambda$ and for seven different values of initial saturation $S_{in}$. For a given value of $\lambda$ and $S_{in}$, the average width and velocity of the four different distributions of the intrinsic permeability is calculated and plotted. With decreasing $\lambda$ (the relative permeability is increasing), the diffusion-like behavior is observed for lower initial saturation, hence the width of the moisture profile is increasing. This is expected because the effect of relative permeability on the flow regime should be similar

to the effect of intrinsic permeability and dynamic viscosity, see Fig. B2a. Note that the effect of relative permeability is more pronounced because the relative permeability varies more significantly for different values of $\lambda$ compared to the sensitivity analysis shown in Fig. B2.

     The velocity of the moisture profiles is increasing with decreasing $\lambda$ for lower initial saturation values. However, this does not apply for initial saturation $S_{in} = 0.02$ and higher. This is because for lower $\lambda$, a diffusion-like behavior is observed for

lower values of initial saturation, hence the moisture profile slows down significantly.

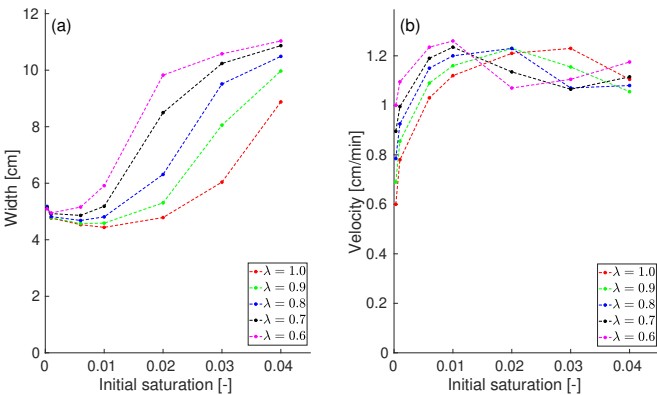

**Figure B3.** The effect of the relative permeability on the flow regime. The width **(a)** and the velocity **(b)** of the finger (or wetting front) at $t = 25$ minutes is plotted against the initial saturation for five different values of parameter $\lambda$. The average of four different distributions of the intrinsic permeability is plotted.

     The Bauters' paradox is again observed for all values of $\lambda$. As $\lambda$ increases, both the minimum width and maximum velocity occur for higher values of initial saturation. This is because the diffusion-like behavior occurs at higher values of initial saturation. The manifestation of the Bauters' paradox is thus shifted to higher initial saturation values. This can also be seen in Fig. B2, but the effect is not so pronounced.

**Effect of the retention curve on the flow regime**

The effect of the retention curve on the flow regime is tested using different parameters $\alpha_w$ and $\alpha_d$ related to the main wetting and main draining branches, respectively. This is done by multiplying the basic values of the parameters $\alpha_w$ and $\alpha_d$ (given in

Table 1) by the free parameter $\alpha$. Both values $\alpha_w$ and $\alpha_d$ are multiplied by the same parameter $\alpha$ ranging from 0.70 to 1.30. Obviously, the baseline simulations are given for $\alpha = 1.00$. The parameters $n_w$ and $n_d$ are fixed and are given in Table 1.

Note that with increasing $\alpha$, the main branches become flatter. This is analogous to using a porous medium with coarser grains. On the other hand, as $\alpha$ decreases, the main branches get steeper, analogous to using a porous medium with finer grains. Figure B4 shows the width and the velocity of the fingers (moisture profiles) 25 minutes from the beginning of infiltration for five different values of $\alpha$ and for seven different values of initial saturation $S_{in}$. For a given value $\alpha$ and $S_{in}$, the average width and velocity of the four different distributions of the intrinsic permeability is calculated and plotted. Again, a snapshot

of the saturation field at 25 minutes for the intrinsic permeability distribution defined by Fig. B1a is shown in the right panel of Fig. B6. As $\alpha$ decreases, the width of the moisture profiles increases rapidly because the diffusion-like behavior is observed for lower values of initial saturation. This is consistent with experimental observations, as the diffusion-like behavior is more readily observed in porous media with finer grains compared to coarser grains; see e.g. the experiments in Cremer et al. (2017). Next, the velocity of the moisture profiles decreases as $\alpha$ decreases. This is expected because at lower values of $\alpha$, the flow

behaves much more diffusion-like and therefore the moisture profiles are slower.

     The Bauters' paradox occurs for all tested values $\alpha$. Moreover, similarly to the effect of the relative permeability, the minimum width and maximum velocity occur for higher initial saturation as $\alpha$ increases, so that the manifestation of the Bauters' paradox is shifted to higher initial saturation values.

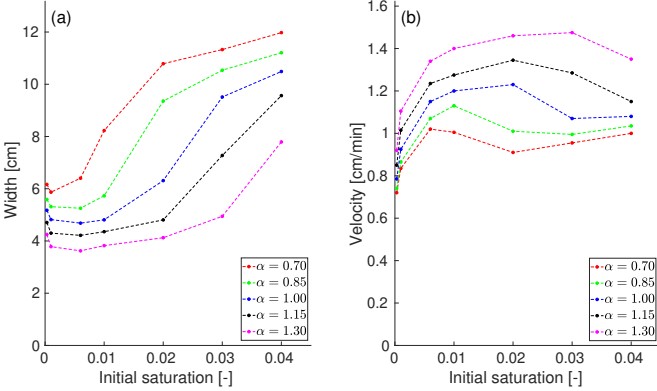

**Figure B4.** The effect of the retention curve on the flow regime. The width **(a)** and the velocity **(b)** of the finger (or wetting front) at $t = 25$ minutes is plotted against the initial saturation for five different values of parameter $\alpha$. The averages of four different distributions of the intrinsic permeability are plotted.

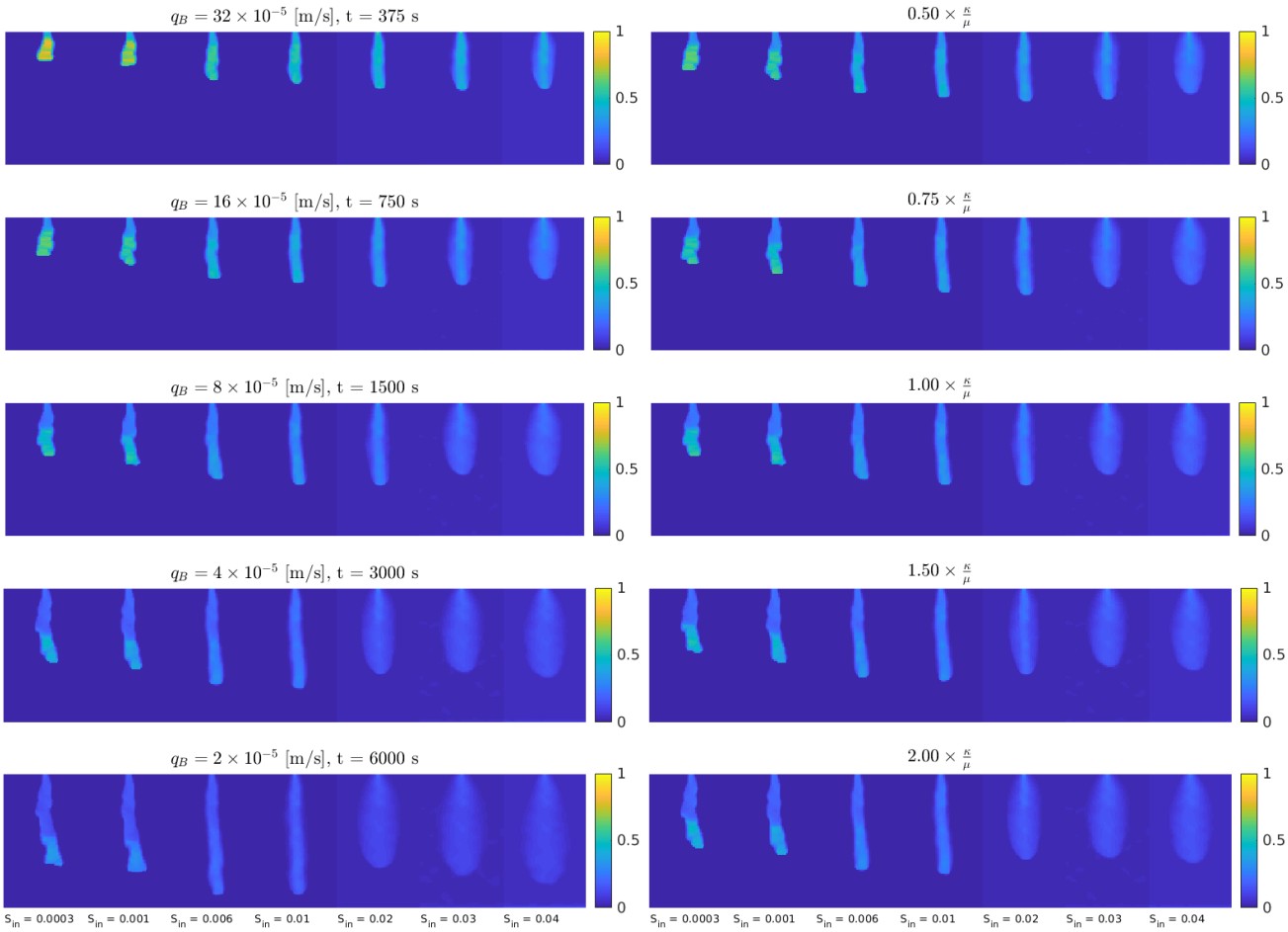

**Figure B5.** Left panel: A snapshot of the saturation field for the intrinsic permeability distribution shown in Fig. B1a for seven different values of initial saturation $S_{in}$ and for five different values of $q_B$ (left panel) and for five different values of $b$ (right panel). For the left panel, times are scaled according to the boundary flux $q_B$, thus a snapshot of the saturation field is shown at $t = 100, 50, 25, 12.5$ and $6.25$ minutes for $q_B = 2, 4, 8, 16, 32 \times 10^{-5}$ ms$^{-1}$, respectively. For the right panel, times are given at 25 minutes.

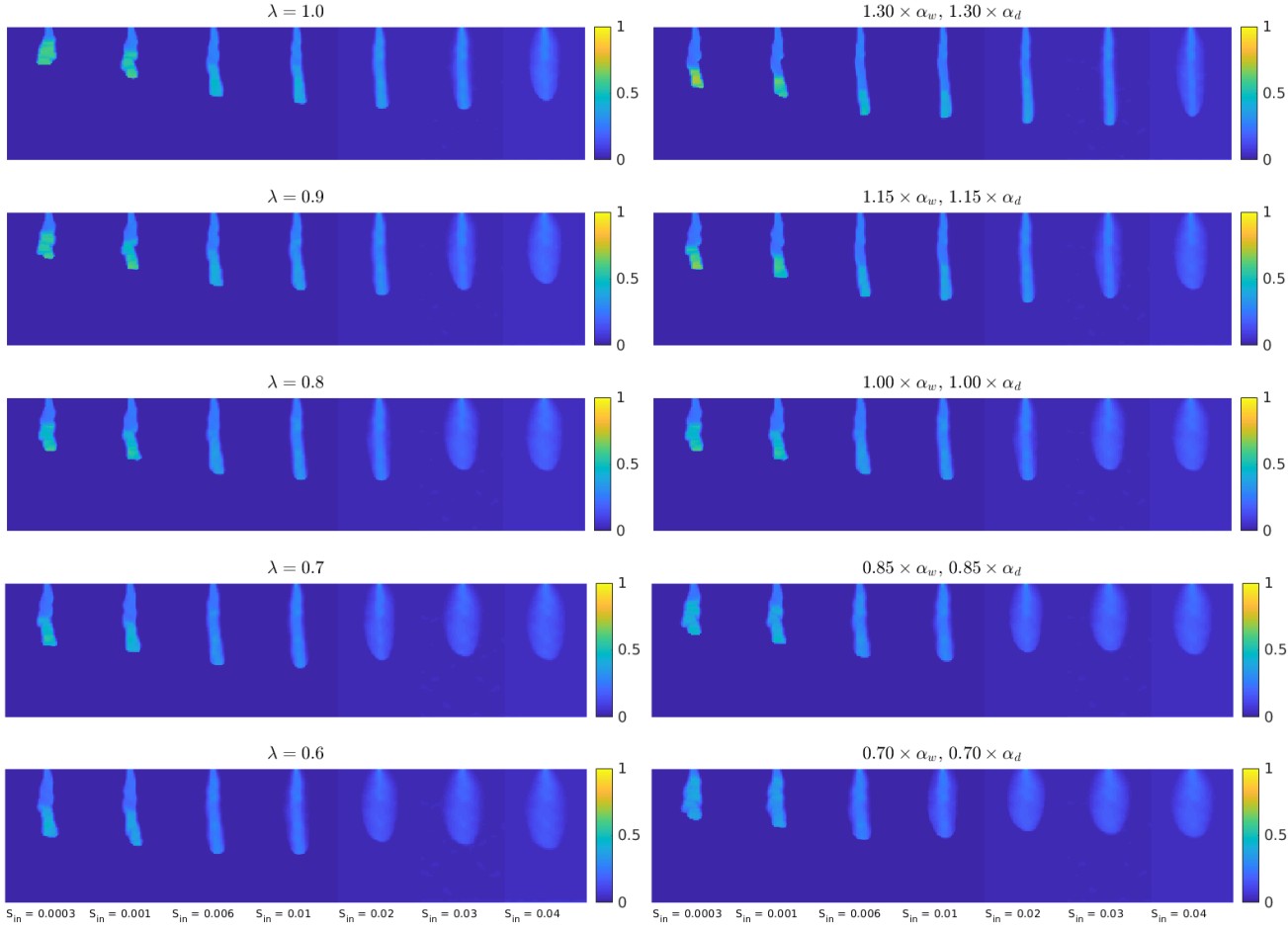

**Figure B6.** A snapshot of the saturation field at 25 minutes for the intrinsic permeability distribution shown in Fig. B1a for seven different values of initial saturation $S_{in}$ and five different values of $\lambda$ (left panel) and for five different values of $\alpha$ (right panel).

*Code and data availability.* The software code that produced the simulations is written in MatLab and can be downloaded from Kmec

(2022). Simulation data that are needed to create the plots included in the manuscript can be downloaded from Kmec et al. (2022). Please do not hesitate to contact us if you encounter any problems when downloading the software code and simulation data.

*Author contributions.* JK and MS wrote the manuscript, TF reviewed the manuscript, TF, MS, RV and JK proposed the model, JK implemented the computer code and ran the simulations, RV checked the mathematics.

*Competing interests.* The authors declare that they have no conflict of interest.

*Acknowledgements.* Jakub Kmec, Tomáš Fürst and Rostislav Vodák gratefully acknowledge the support by the Operational Programme Research, Development and Education, project no. CZ.02.1.01/0.0/0.0/17_049/0008422 of the Ministry of Education, Youth and Sports of the Czech Republic. Rostislav Vodák was supported by the Ministry of Education, Youth and Sports of the Czech Republic, project no. CZ.02.1.01/0.0/0.0/17_049/0008408 Hydrodynamic Design of Pumps. Computational resources were supplied by the project "e-Infrastruktura CZ" (e-INFRA LM2018140) provided within the program Projects of Large Research, Development and Innovations Infrastructures. Tomáš

Fürst gratefully acknowledges the generous support of the Fulbright Commission in scope of the Masaryk-Fulbright Scholarship.

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
