# Peer review of "Semi-continuum modelling of unsaturated porous media flow to explain the Bauters' paradox"

_EGUsphere, 2022_

## Referee Comment (RC3)

The paper presents modeling results for infiltration into porous media with low initial saturation and addresses the representation of water fingers with an overshoot at the tip that has been discussed intensely in the literature in the last years. The focus is here on a reproduction of the finger width depending on the initial saturation and the related finger speed from experimental results from the literature. The important point is to capture the non-monotonic relationship found in the experiments. The model is a space discretized mass balance and it includes hysteretic capillary pressure saturation relations. The topic is of interest for the journal. However, I think that a lot of clarification is needed. I find it a great achievement that experiments and in particular the dependency of finger width on initial saturation could be reproduced with a model. But the paper makes the point that this is due to a novel model approach that is used here. The model has been published in previous papers and the ideas behind it are here not explained. However, as the claim is made that the key to reproduce the experimental findings is the model approach that is different from previous ones, one cannot follow the reasoning. Although it is understandable that one does not want to repeat too much previous work, enough explanation is needed to make it possible to follow the lines of argumentation.

- The biggest problem I have is that it is unclear to me, how the model (not in its concept but simply the set of equation that is solved in the end) differs from the discretized Richards equation solution with a specific hysteretic capillary pressure saturation relation. The authors stress in line 292 that the model is not a numerical scheme to solve the Richards equation. But they do not explain what the difference is and what could not be reproduced with a discretized Richards equation. Also, it is written that the choice of the grid blocks has physical interpretation, but they do not give this interpretation. It is referred to a previous paper (Vodak et al., 2022), but the main story line of a paper should be understandable without reading further papers.

  The difference between this model and a discretized Richards equation does not become clear to me. I understand that the choice of capillary pressure curve is made dependent on the grid size (or block size, without explanation of the concept it is not so clear why this should be different) and that this is somehow related to not covering an REV with the grid size. This is called scaling. But apart from the reasoning, one chooses in the end a (hysteretic) capillary pressure curve. This leads to the same system of equations that one would obtain with a discretized Richards equation with a specific choice of parameters. Eqs. (1) and (4) combined with a capillary pressure saturation function would be the same equations one would solve if one discretizes the Richards equation with a standard finite volume, two-point flux approximation scheme and explicit Euler time discretization. The saturation would be represented exactly as is outlined in lines 67-71.

  The authors use a hysteretic capillary pressure saturation function and the shape depends on the grid size. With decreasing grid size, the chosen curve gets flatter. They stress that the choice of the grid size is not trivial, but this is not further outlined. In the end, one fixes a grid cell and with this one chooses a capillary pressure saturation function. Whatever the reasoning behind this choice is, the function is fix. It would not make a difference if the function would have been chosen with a different reasoning. Finally, one solves the same system of equations as one

would if one solves the Richards equation with a finite volume scheme using the same hysteretic capillary pressure function. One would also generate the same result. So the unstable infiltration and the finger width behaviour would be reproducible with the standard Richards equation using the same hysteretic capillary pressure function that has been used here. No new model is needed for this. I guess the key point is thus the hysteretic capillary pressure function and the choice of the function that depends on grid size. I assume there is more to the choice of the capillary pressure function and that one could predict is from material properties and knowledge of the flow regime. That would make a difference, because one could then say that this model is predictive, while the discretized Richards equation is not. But this is only guessing.

- It is claimed that using the geometric mean of the cell permeabilities to approximate the flux across volume interfaces is usually not done, but at this point I disagree. Often one uses the geometric or the harmonic mean of the saturated permeabilities and upstream weighting of the relative permeabilities, but to my knowledge it is as common to use geometric means of the total permeability. In textbooks on numerical solutions of two-phase flow equations all these options are usually discussed. It is interesting though, that the geometric mean is a key element to reproduce the fingers.

- The authors argue that the model converges to a new type of model if $\Delta x$ goes to zero. If I understand correctly, this goes with the changing capillary pressure saturation curve, which goes from steep to flat with decreasing size $\Delta x$. As the authors write in line 278, the model should be able to reproduce the standard Richards equation behaviour. I do not see how the drainage from a fully saturated soil column towards a hydrostatic profile should be reproducible with this model as $\Delta x$ goes to zero. The hydrostatic saturation profile should match the primary drainage capillary pressure curve. If the curve gets flat with decreasing $\Delta x$, one would not be able to retrieve the profile. One would get a sharp change from fully saturated to dry.

- I understand that the authors want to acknowledge the experimental findings of Bauters et al., 2000, and to highlight their achievements by calling the nonmonotonic dependency of finger width with initial saturation Bauters' paradoxon. Still, I find this wording a bit odd. A paradoxon involves a self-contradicting aspect or something that is against the intuitive expectation. This should not diminish the observations, and there is maybe not an easy explanation, but I find it hard to see a paradoxon.

- It is not clear if the model results presented here are predictive or if parameter and other adjustment was involved. In line 245 it is written that a four times lower infiltration rate than in the experiments was used. Why was this lower infiltration rate chosen? Was the match with the experiments not obtained with the same infiltration rate as in the experiments? I find this an important point, as a model needs to be predictive, meaning that one should be able to know the parameters from information about the materials or from measurements of the materials. It is an achievement to reproduce non-monotonic finger width with initial saturation,

but if this was obtained with a model that needed fitting, one could argue that one would have obtained the same with a classical (hysteretic) Richards equation model by fitting the capillary pressure curve.

- The discussion on the REV in lines 42-51 is a bit long. The problem of the REV for two-phase flow problems has been discussed a lot (for example already in the papers on volume averaging of two-phase flow) and it is acknowledged that the REV for fluid content is problematic, in particular for unstable displacement. The question of an REV for pressure in porous media has also been discussed in the literature (just one example: Nordbotten et al., Water Resources Research 2008). I think this could be shortened and does not need all the citations.

- Line 56: I think that at least the papers of Lenormand et al., 1983, or Wilkinson, 1986, are here misleading. The point in these papers is not to derive alternative models to the Richards equation but to capture viscous and capillary unstable immiscible displacement. To my knowledge, they do not include gravity.

- Line 81: This new type of mathematical models of Nature sounds a bit overselling. The switch from parabolic and hyperbolic for two-phase flow problems is known for continuum models in the limit that capillary effects vanish (also for immiscible displacement in the fractional flow formulation).

- Line 237: Where does the 0.0005 come from? It is in none of the figures.

- Line 242-243: I think this is a bit simplifying. The solutions of the Richards equation without hysteresis are stable, so of course this effect is not captured.

---

## Author Comment (AC1)

**Point to point response to reviewer RC1**

**Citation of the review: https://doi.org/10.5194/egusphere-2022-673-RC1**

We thank the reviewer for helpful and constructive comments. Below you will find detailed responses to all the reviewer's comments. The reviewer's comments are highlighted in red, our comments are in black.

The manuscript describes a modeling approach of the flow in unsaturated sand in order to explain so-called Bauters' paradox. A semi-continuum model approach is presented that can imitate the finger flow phenomena with the oversaturation in the tip and the whole transition towards a diffusion-type infiltration plume as function of the initial soil moisture condition. The model was successfully tested by description of the Bauters' experiment.

**General Comments:** The manuscript tackles an important problem that is in the focus of the journal. Methods and results are novel and can much contribute to progress in describing unsaturated flow phenomena.

**Major issue 1:** My main critical comment is that for in-depth understanding the physical basis of the modeling approach and finally also of the results, a better explanation of the semi-continuum's model concept would help.

A detailed explanation of the concept of the semi-continuum model is published in [1]. However, we agree with the reviewer that a better explanation of the semi-continuum model will improve the manuscript. Since the reviewer's comment (**major issue 3**) addresses the same issue, we refer to the explanation in the response to **major issue 3**.

**Major issue 2:** In addition, I found the introduction is much too far reaching, of course, everything is somehow connected. The review on global water cycle, the flow phenomena, sample volume dependency, REV concepts and Richards' equations, and other aspects, all does not help much to better understand the problem, and are not discussed later any more.

We will update the introduction. First, the review of global water cycle is not relevant for the main point of the manuscript, and therefore the first paragraph in the Introduction will be replaced by the following text: "Infiltration of rainwater into soil forms an essential part of the hydrological cycle. Therefore, research on the movement of water in soil has long been a focus of attention. The origins of infiltration research were substantially influenced by the idea to describe the movement of water in soil by diffusion-like models [2]. Later, it was discovered that – even in homogeneous porous materials – flow may become spatially very inhomogeneous. Most of the infiltrating water flows through preferential pathways (fingers) leaving islands of dry material behind. In the fingers, the so-called saturation overshoot often occurs – the finger tip becomes much more saturated than the finger tail. This type of flow cannot be described by the diffusion-like models [3]. However, it is well described by a semi-continuum model introduced in [1]. In this paper, we demonstrate that this semi-continuum model captures infiltration into an unsaturated homogeneous porous medium comprehensively, in the sense that it correctly describes the experimentally observed complicated transition between the finger-like and diffusion-like flow regimes."

Second, sample volume dependency, the concept of REV, Richards' equation, etc. are closely related to the semi-continuum model, specifically the scaling of the retention curve. This will be better explained in the manuscript and the introduction will be improved accordingly. Please, see the response to **major issue 3**, where this is explained in detail.

**Major issue 3:** What I did not understand was the semi-continuum model concept, especially what is different from a numerical discretization of a continuum model? Perhaps you did it already in other papers. It seems relatively simple and more empirical because of the block size selection and the scaling relations. Maybe it helps to include an illustration? The idea of scaling the retention function with block size is also unclear to me. I did not read the cited references, but the present paper seems to apply the previously developed model concept to describe the specific experiment, yes?

Let us first answer the last question: *I did not read the cited references, but the present paper seems to apply the previously developed model concept to describe the specific experiment, yes?* Indeed, this paper uses a previously developed semi-continuum model to describe the specific experiments of Bauters et al. [4]. For clarity, this statement will be included in the manuscript. We chose the experimental results of Bauters et al. because no other model has been able to reproduce them so far.

Another comment by the reviewer relates to understanding the semi-continuum model concept that is closely related with the scaling of the retention curve (see Figure 1 in the manuscript). Understanding this concept may not be clear, so we provide a brief discussion below that we plan to include in the manuscript. However, the detailed mathematical and physical justification is already published in [1], hence for a deeper understanding we refer to this paper.

[revised manuscript text omitted]

**Major issue 4:** It is suggested in the manuscript that the initial water saturation is the variable controlling the finger formation. What about the wettability (i.e., surface tension), which is of course connected with water saturation but can change with time?

The experiments by Bauters et al. [4] that we have reproduced in the manuscript were carried out on pure quartz sand. This sand does not change its wettability according to the duration of contact with distilled water.

Therefore, we did not consider the effect of the change in wettability on the flow of water through the sand because this effect is not the cause of the Bauters' paradox.

Overall, this valuable contribution could become even better if more focused and with more specific explanations on the physical basis of the approaches.

Thank you!

---

## Author Comment (AC2)

**Point to point response to reviewer RC2**

**Citation of the review: https://doi.org/10.5194/egusphere-2022-673-RC2**

We thank the reviewer for the valuable feedback and constructive comments. Below you will find the detailed responses to all the reviewer's comments. The reviewer's comments are highlighted in red, our comments are in black.

The paper analyses the process of infiltration of a liquid into an unsaturated porous medium, which is characterized by different regimes depending on the level of saturation. The liquid is assumed to be wettable with respect to the homogeneous medium. Then, the authors introduce a semi-continuum model that lends itself well to the interpretation of the infiltration regime.

The topic is of interest and deals with a process that even in extremely simple situations reveals a degree of complexity.

**Major issue 1:** However, I see the major limitation in the absence of experimental validation: the numerical scheme, although of interest, appears limited in its interpretation of physical reality.

We agree that a more detailed explanation of the semi-continuum model should be included in the manuscript to make the manuscript easier to understand. Since reviewer RC1 raised a similar comment, we copied the response to the reviewer RC1 below.

The concept of the semi-continuum model is closely related with the scaling of the retention curve (see Figure 1 in the manuscript). Understanding this concept may not be clear, so we provide a brief discussion below that we plan to include in the manuscript. However, the detailed mathematical and physical justification is already published in [1], hence for a deeper understanding we refer to this paper.

[revised manuscript text omitted]

**Major issue 2:** Moreover, and this is a second major issue, there is a lack of sensitivity analysis, since the only estimator comes from repeating the numerical tests by changing only the intrinsic permeability distribution. In this sense, it is necessary for the authors to thoroughly analyse the uncertainty of the parameters (viscosity, degree of saturation, etc.) and analyse the variability of the governed quantities.

We agree and provide the sensitivity analysis. All the simulations are computationally demanding, and so distributed computing infrastructure "e-Infrastruktura CZ" (e-INFRA LM2018140) has been used. The sensitivity analysis was performed by varying intrinsic permeability, dynamic viscosity, relative permeability, retention curve and the boundary flux. This was done for four different distributions of the intrinsic permeability and seven different values of initial saturation. In total, 588 simulations were performed. However, for the block size that is used in the manuscript ($\Delta x = 0.25$ cm), a single simulation takes almost 30 days. Thus, the sensitivity analysis could not be performed for $\Delta x = 0.25$ cm. Therefore, a larger block size $\Delta x = 0.50$ cm was used while the rest of parameters remained the same as in the manuscript. Let us note that even with the larger block size, all the simulations took almost 1000 CPU days.

The results of the sensitivity analysis are presented below. It is demonstrated that the Bauters' paradox occurs for different values of material parameters or different boundary conditions. Some figures related to the sensitivity analysis are included at the end of this response so that this text is not too extensive. We suggest to include these figures in the appendix of the manuscript.

**Effect of the intrinsic permeability and dynamic viscosity on the flow regime**

Increasing the intrinsic permeability $\kappa$ has the same effect as decreasing the parameter $\mu$ and vice versa. Therefore, a fraction $\frac{\kappa}{\mu}$ is used for the sensitivity analysis of both these parameters. The baseline values of $\kappa$ and $\mu$ are given in Table 1 in the manuscript. Five different values $b \cdot \frac{\kappa}{\mu}$ were examined, where $b = 0.50, 0.75, 1.00, 1.50$ and $2.00$. Obviously, baseline simulations are given for $b = 1.00$. For each value of $b$, 28 different simulations are performed, with variable initial saturation (seven different initial saturation) and variable intrinsic permeability distribution (four different distributions). To generate spatially correlated distribution of the intrinsic permeability, the same approach was used as in the manuscript; see Fig. A.1 at the end of this response. Note that 140 different simulations were performed in total. The same scheme was applied to all other sensitivity analysis simulations, i.e. the analysis was always performed for seven different values of initial saturation and four different distributions of intrinsic permeability.

Figure 1 shows the width and the velocity of the fingers (moisture profiles) 25 minutes from the beginning of infiltration for five different values of $b$ and for seven different values of initial saturation $S_{in}$. The width of each moisture profile was calculated in the same way as in the manuscript. 
[revised manuscript text omitted]

Finally, sensitivity analysis for the boundary flux $q_B$ is performed. Five different values of the boundary flux were used ranging from $2 \times 10^{-5}$ ms$^{-1}$ to $16 \times 10^{-5}$ ms$^{-1}$. The baseline simulations are given for $q_B = 8 \times 10^{-5}$ ms$^{-1}$. Figure 4 shows the width and the velocity of the fingers (moisture profiles) for five different values of boundary flux $q_B$ and for seven different values of the initial saturation $S_{in}$. For a given value $q_B$ and $S_{in}$, the average width and velocity of the four different distributions of the intrinsic permeability is calculated and plotted. Snapshot of the saturation field for the intrinsic permeability distribution defined by Fig. A.1**(a)** is shown in Fig. A.5 at the end of this response. Since the used boundary fluxes varied by more than one order of magnitude, the times for which the velocity and the width are calculated need to be scaled according to the boundary flux. The time-points used were $t = 100, 50, 25, 12.5$ and $6.25$ minutes for $q_B = 2, 4, 8, 16, 32 \times 10^{-5}$ ms$^{-1}$, respectively. It can be seen in Fig. 4**(a)** that with decreasing boundary flux, the flow tends to become more diffusive. This is in good agreement with the experimental observation [11]. Moreover, it is not surprising that the velocity of the moisture profiles shown in Fig. 4**(b)** decreases with decreasing boundary flux.

The Bauters' paradox is again observed for all tested values of $q_B$, so that the manifestation of the Bauters' paradox shifts towards higher initial saturation values as the boundary flux increases.

[Figure]

Figure 4: The effect of the boundary flux on the flow regime. The width **(a)** and the velocity **(b)** of the finger (or wetting front) is plotted against the initial saturation for five different values of boundary flux $q_B$. Times are scaled according to the boundary flux $q_B$, thus the width and velocity of the moisture profiles are calculated at $t = 100, 50, 25, 12.5$ and $6.25$ minutes for $q_B = 2, 4, 8, 16, 32 \times 10^{-5}$ ms$^{-1}$, respectively. The averages of four different distributions of the intrinsic permeability are plotted.

**Major issue 3:** There is one point of particular interest: if Bauter's paradox occurs even when intrinsic permeability is homogeneous, how does the proposed model behave in this situation?

As stated in the text of the manuscript (lines 214–216), the Bauters' paradox with constant intrinsic permeability can be observed in Fig. 4.3 in the Dissertation Thesis [12]. However, more physical-looking fingers evolve if the distribution of the intrinsic permeability is included. We believe that including the simulations in the Appendix of the manuscript for the case of homogeneous intrinsic permeability is more appropriate than referring to the Dissertation. Therefore, the same simulations as in the manuscript have been performed with homogeneous intrinsic permeability. The parameters used for simulations are given in Table 1 in the manuscript. Figure 5 shows a snapshot of the saturation field at 25 minutes for seven different values of the initial saturation. It is observed that the Bauters' paradox occurs even in this case. The effect of the intrinsic permeability distribution is pronounced for the initially dry porous medium, while for the initially wet porous medium this effect is negligible. This is expected because in the case of diffusion-like regime, small changes in intrinsic permeability do not have a significant effect on the flow. The artificial looking behavior for the initially dry porous medium is eliminated if a more realistic porous medium is used for the simulations, i.e. if the distribution of the intrinsic permeability is included.

[Figure]

Figure 5: Snapshot of the saturation field at 25 minutes for seven different values of the initial saturation. The distribution of the intrinsic permeability is not included, i.e. the medium is perfectly homogeneous. Saturation values are colour-coded according to the colour bar on the right. Initial saturation of the medium increases from left to right.

**Major issue 4:** Last, the authors tell us about the behavior of their model, but the interpretation of why appears to be missing.

We agree with the reviewer that interpretation/explanation why the Bauters' paradox occurs is missing. We suggest to include the following explanation in the manuscript.

We conjecture that the explanation of the Bauters' paradox is rather similar to the non-monotonic dependence of porous medium flow on the magnitude of the influx. For very small values of influx, the flow becomes stable with increasing finger width. The same applies for very large values of influx. Hence, the unstable flow is only observed for fluxes within a certain range [13, 14, 15]. Yao and Hendrickx [13] hypothesized that the stable flow occurs when the effect of gravity becomes negligible. This happens in two "extreme" cases. First, at very low infiltration rates, capillarity becomes the dominant force compared to the force of gravity. Second, for infiltration rates higher than the saturated hydraulic conductivity, the viscosity dominates and the stable flow without fingers occurs. In our case, the dependence of the flow regime on the initial saturation behaves similarly. For initially dry porous medium, the capillarity dominates and the large capillary forces are able to win over gravity in sucking the water sideways into dry areas of the matrix. In a medium, which is moderately wet, this becomes more difficult, because the capillary forces are generally lower. Thus, in a moderately wet medium, the fingers become thinner and faster. At sufficiently high initial saturation, the large conductivity between neighboring blocks prevents water piling up behind the wetting front and the formation of saturation overshoot. This results in the ability of lateral expansion because the persistence of the fingers is suppressed [16, 17]. Therefore, a diffusion-like flow regime is observed.

**Minor comments:**

1. Figure 3: specify units for the colorbar

2. Figures 5-6: add the frame to panel b diagram

These minor issues will be fixed in the manuscript.

**A    Figures**

[Figure]

Figure A.1:    The distribution of the intrinsic permeability.  The distributions satisfy:  **(a)** $\kappa_{max}/\kappa_{min} \approx 3.40$, **(b)** $\kappa_{max}/\kappa_{min} \approx 3.48$, **(c)** $\kappa_{max}/\kappa_{min} \approx 2.28$, **(d)** $\kappa_{max}/\kappa_{min} \approx 2.42$.  Intrinsic permeability values are colour-coded according to the colour bar on the right.

[Figure]

Figure A.2: A snapshot of the saturation field at 25 minutes for the intrinsic permeability distribution shown in Fig. A.1(a) for five different values of $b$ and for seven different values of initial saturation $S_{in}$.

[Figure]

Figure A.3: A snapshot of the saturation field at 25 minutes for the intrinsic permeability distribution shown in Fig. A.1(a) for five different values of $\lambda$ and for seven different values of initial saturation $S_{in}$.

[Figure]

Figure A.4: A snapshot of the saturation field at 25 minutes for the intrinsic permeability distribution shown in Fig. A.1(a) for five different values of $\alpha$ and for seven different values of initial saturation $S_{in}$.

[Figure]

Figure A.5: A snapshot of the saturation field for the intrinsic permeability distribution shown in Fig. A.1(a) for five different values of $q_B$ and for seven different values of initial saturation $S_{in}$. Times are scaled according to the boundary flux $q_B$, thus a snapshot of the saturation field is shown at $t = 100, 50, 25, 12.5$ and $6.25$ minutes for $q_B = 2, 4, 8, 16, 32 \times 10^{-5}$ ms$^{-1}$, respectively.

---

## Author Comment (AC3)

**Point to point response to reviewer RC3**

**Citation of the review: https://doi.org/10.5194/egusphere-2022-673-RC3**

We thank the reviewer for useful and meaningful comments. Below you will find detailed responses to all the reviewer's comments. The reviewer's comments are highlighted in red, our comments are in black.

The paper presents modeling results for infiltration into porous media with low initial saturation and addresses the representation of water fingers with an overshoot at the tip that has been discussed intensely in the literature in the last years. The focus is here on a reproduction of the finger width depending on the initial saturation and the related finger speed from experimental results from the literature. The important point is to capture the non-monotonic relationship found in the experiments. The model is a space discretized mass balance and it includes hysteretic capillary pressure saturation relations. The topic is of interest for the journal. However, I think that a lot of clarification is needed. I find it a great achievement that experiments and in particular the dependency of finger width on initial saturation could be reproduced with a model. But the paper makes the point that this is due to a novel model approach that is used here. The model has been published in previous papers and the ideas behind it are here not explained. However, as the claim is made that the key to reproduce the experimental findings is the model approach that is different from previous ones, one cannot follow the reasoning. Although it is understandable that one does not want to repeat too much previous work, enough explanation is needed to make it possible to follow the lines of argumentation.

We appreciate valid and positive feedback. Let us note that we indeed did not want to repeat too much the previous work of Vodák et al. [1], where the detailed mathematical and physical justification is published. However, we agree that for better clarity, it is necessary to explain the semi-continuum model in more details. The most important is the first reviewer's comment below. This comment is primarily related to understanding the semi-continuum model and how it differs from the Richards' Equation (RE). Since reviewer RC1 and reviewer RC2 raised a similar comment, we copy the response to them first that we plan to include in the manuscript.

"The scaling of the retention curve, i.e. the dependence of the capillary pressure-saturation relation on the block size, is not a common approach in flow modelling. However, the dependence of the experimentally determined retention curve on the porous medium sample size has been observed for a long time [2, 3, 4, 5, 6, 7]. The concept of REV is essential in this case because if the sample of porous medium is smaller than REV, key physical quantities, such as the retention curve, are strongly dependent on the sample size. The crucial idea of the semi-continuum model is to include this dependency in the model, i.e. to scale the retention curve according to the block size. In the semi-continuum model, a block represents a real sample of the porous material. This makes the semi-continuum model fundamentally different from numerical schemes for solving partial differential equations where the block plays only a discretization (i.e. mathematical) role and regardless of the block size, the retention curve remains the same. In the semi-continuum model, the computational mesh (the blocks) takes into account the dependence of the physical parameters on the size of the blocks. Surprisingly, the idea of taking REV size into account in modelling porous media has been around for a long time. For instance, in [8], the authors estimated the size of the REV and used it as a lower limit for the size of the finite elements. They argue that the use of smaller elements would lead to violation of continuum assumptions and thus the continuum approximation would no longer be appropriate. The same idea is used in the semi-continuum model: For blocks smaller than the REV, scaling of the retention curve must be included because the continuum approximation is no longer adequate. Because we are interested in the description of flow phenomena below the REV scale, we need to include the dependence of the retention curve on the block size. This scaling of the retention curve must meet a physically justified requirement that the nature of the flow is preserved across all levels of block size. This means that the fluxes between neighboring blocks must not change when $\Delta x$ changes. Given equation (4) in the manuscript, if $\Delta x$ decreases by half, the fluxes increase by a factor of two if the scaling of the retention curve is not included. Therefore, a linear scaling of the retention curve is introduced in equation (6) in the manuscript, so the fluxes between blocks remain the same as $\Delta x$ decreases. For

more details, see figures Fig. 4–6 in [1] that show the numerical convergence of the semi-continuum model in 1D and 2D.

The natural question is what the limit of the semi-continuum model would be as $\Delta x \to 0$. We tried to answer this question in [1] and derived the limit equation in a single spatial dimension:

$$(K_{PS}\partial_t S - \partial_t P_H)(P_H - v) \geq 0, \qquad \text{for all } v \in [C_2, C_1], \qquad \text{and } P_H \in [C_2, C_1], \tag{1}$$

$$\theta \partial_t S + \partial_x \left( \frac{\kappa}{\mu}\sqrt{k(S^-)}\sqrt{k(S^+)}(\rho g - \partial_x P_H) \right) = 0, \qquad S^\pm(x_0, t) = \lim_{x \to x_0^\pm} S(x, t). \tag{2}$$

In this equation, $\kappa$ denotes the intrinsic permeability, $\rho$ the fluid density, $g$ acceleration due to gravity, $\mu$ the dynamic viscosity of fluid, and $S$ the saturation. The values $C_1$ [Pa] and $C_2$ [Pa] denote the constant limits of the main wetting and draining branches, respectively. The limit is a partial differential equation containing a Prandtl-type hysteresis operator $P_H$ under the space derivative. If we are located on the main wetting or draining branches, the limit equation becomes a hyperbolic differential equation. Between the two main branches (i.e., we are located on the scanning curve), the limit represents a parabolic differential equation. It means the limit switches between parabolic and hyperbolic types of equation. The limit equation is a new type of mathematical model – we are not aware of any research that has investigated equations of this type. Note that the Richards' equation is a parabolic type equation – that is why it is only able to simulate the diffusion-like flow regime [9]."

1. The biggest problem I have is that it is unclear to me, how the model (not in its concept but simply the set of equation that is solved in the end) differs from the discretized Richards equation solution with a specific hysteretic capillary pressure saturation relation. The authors stress in line 292 that the model is not a numerical scheme to solve the Richards equation. But they do not explain what the difference is and what could not be reproduced with a discretized Richards equation. Also, it is written that the choice of the grid blocks has physical interpretation, but they do not give this interpretation. It is referred to a previous paper (Vodak et al., 2022), but the main story line of a paper should be understandable without reading further papers.

The crucial difference between the Richards' Equation (RE) and the semi-continuum model is the scaling of the retention curve. The semi-continuum model is not a discretized RE with a specific hysteretic capillary pressure saturation relation. The RE uses a fixed retention curve as $\Delta x$ decreases, so for $\Delta x \to 0$ there is only diffusion-like behavior. On the contrary, the semi-continuum model scales the retention curve according to the size of $\Delta x$. This linear scaling might seem like a small trick, but when a mathematical analysis of the semi-continuum model is done for $\Delta x \to 0$, it turns out that the limit of the semi-continuum model is significantly different from the RE, as can be seen from equations (1) and (2). The limit switches between a parabolic and hyperbolic differential equation for unsaturated porous medium, while the RE is a parabolic differential equation. And it is the hyperbolic behavior of the semi-continuum model that makes it possible to model overshoot and all that it implies.

For clarity, let us show also a numerical analysis of the semi-continuum model and the RE in one dimension for $\Delta x \to 0$. The numerical analysis is shown in Figure 1, which is reprinted from Vodák et al. [1] (specifically, Figure 1 is here composed of figures 2 and 4 from [1]). Left panel of Figure 1 shows a convergence of the semi-continuum model, i.e. the scaling of the retention curve is included. Convergence of the moisture profile is shown at $t = 10$ minutes for $\Delta x \to 0$ for initial saturation $S_{in} = 0.01$, and constant top boundary flux $q_0 = 6 \times 10^{-5}$ m/s. The scaling of the retention curve preserves the character of the flow across all levels of $\Delta x$. Moreover, the moisture profile obviously converges and retains the overshoot pattern. Right panel of Figure 1 shows the same convergence of the moisture profile as in the left panel of Figure 1, but the scaling of the retention curve is not included. This can be considered as a numerical scheme of the RE. As we can see, without proper scaling of the retention curve, the overshoot behavior would disappear. Therefore, the semi-continuum model and RE differs significantly as $\Delta x$ decreases.

According to the physical interpretation of the blocks, please see the discussion we already included above the reviewer's comment number 1. As we mentioned, we plan to include this discussion in the manuscript. There is also discussed the role of the blocks. For the sake of clarity, we also provide the answer here: "In the semi-continuum model, a block represents a real sample of the porous material. This makes the semi-continuum model fundamentally different from numerical schemes for solving partial differential equations where the block plays only a discretization (i.e. mathematical) role and regardless of the block size, the retention curve remains the same. In the semi-continuum model, the computational mesh (the blocks) takes into account the dependence of the physical parameters on the size of the blocks."

[Figure]

[Figure]

Figure 1: Left panel: The scaling of the retention curve **is** included. Right panel: The scaling of the retention curve **is not** included.

The difference between this model and a discretized Richards equation does not become clear to me. I understand that the choice of capillary pressure curve is made dependent on the grid size (or block size, without explanation of the concept it is not so clear why this should be different) and that this is somehow related to not covering an REV with the grid size. This is called scaling. But apart from the reasoning, one chooses in the end a (hysteretic) capillary pressure curve. This leads to the same system of equations that one would obtain with a discretized Richards equation with a specific choice of parameters. Eqs. (1) and (4) combined with a capillary pressure saturation function would be the same equations one would solve if one discretizes the Richards equation with a standard finite volume, two-point flux approximation scheme and explicit Euler time discretization. The saturation would be represented exactly as is outlined in lines 67-71.

It is important to note that in the case of nondecreasing boundary flux, the solution of the RE is nondecreasing point wise, so it never leaves the main wetting branch of the retention curve [9]. Therefore, the solution is stable regardless of whether hysteresis is included because the hysteresis of the retention curve never comes into action. In the case of the experiment by Bauters et al. [10] the authors used the constant boundary flux, but the saturation profile was not stable for lower initial saturation. This means that the solution of the RE cannot capture the complex behavior of this experiment, as the RE is always stable in this case. This is also consistent with our numerical results, because if the scaling of the retention curve is not included, the solution becomes stable as $\Delta x$ decreases, see the right panel of Figure 1.

For a given $\Delta x$ and retention curve, the semi-continuum model may indeed look like a numerical scheme for the RE. However, when using different $\Delta x$, a different retention curve must be used for the RE. For the semi-continuum model, we define the retention curve for the reference block size $\Delta x_0$ and the retention curve is then scaled linearly with the block size. Note also that if a partial differential equation provides a certain solution effect (for example, the RE is incapable of producing any overshoot for monotonic influx boundary conditions), but its so-called "numerical scheme" provides the opposite effect, then it cannot be its numerical scheme. Since the semi-continuum model (i.e., the scaling of the retention curve is included) produces the overshoot for the monotonic influx boundary condition, then it is not a numerical scheme of the RE. And as we have already mentioned above, as $\Delta x$ decreases, the semi-continuum model does not converge to the RE (see equations (1), (2) and Fig. 1).

The authors use a hysteretic capillary pressure saturation function and the shape depends on the grid size. With decreasing grid size, the chosen curve gets flatter. They stress that the choice of the grid size is not trivial, but this is not further outlined. In the end, one fixes a grid cell and with this one chooses a capillary pressure saturation function. Whatever the reasoning behind this choice is, the function is fix. It would not make a difference if the function would have been chosen with a different reasoning. Finally, one solves the same system of equations as one would if one solves the Richards equation with a finite volume scheme using the same hysteretic capillary pressure function. One would also generate the same result. So the unstable infiltration and the finger width behaviour would be reproducible with the standard Richards equation using the same hysteretic capillary pressure function that has been used here. No new model is needed for this. I guess the key point is thus the hysteretic capillary pressure function and the choice of the function that depends on grid size. I assume there is more to the choice of the capillary pressure function and that one

could predict is from material properties and knowledge of the flow regime. That would make a difference, because one could then say that this model is predictive, while the discretized Richards equation is not. But this is only guessing.

This is a good point. Indeed, the capillary pressure function is a material characteristic and can be predicted from material properties and knowledge of the flow regime. This makes the semi-continuum model predictive. Without this, of course, such a model would not be appropriate for any application. We plan to include a new paragraph on this issue in the discussion section:

"Note that for a given $\Delta x$ and retention curve, the semi-continuum model may look like a numerical scheme for the RE. However, when using different block size $\Delta x$, a new retention curve must be used for the RE to retain the character of the flow. Otherwise, only the diffusion-like behavior occurs [9]. In contrast, for the semi-continuum model, we define the retention curve for the reference block size $\Delta x_0$ and the retention curve is then scaled accordingly to the block size. In this case, the retention curve is a material characteristic and can be predicted from material properties and knowledge of the flow regime. The semi-continuum model is thus predictive; we do not need to fit the retention curve for each $\Delta x$ separately."

Let us also note that the size of the block is not fitted to achieve the best agreement with the experiments. The adjustment of reference block size $\Delta x_0$ was done independently of the final size of the used block $\Delta x$ (see section 3.1 in the manuscript). The parameter $\Delta x_0$ was calibrated using the block size $\Delta x = 0.50$ cm. The smallest possible block size $\Delta x = 0.25$ cm was then chosen for the simulations to ensure that the simulations were still computable. When using a twice smaller block $\Delta x = 0.125$ cm, a single simulation takes hundreds of days. And there are 80 simulations in total in the manuscript (10 simulations for each intrinsic permeability distribution). Moreover, a good agreement with the experiments is also obtained for larger blocks. For more details, we refer to the response to reviewer's comment number 5, where we show that the model is not fitted to achieve the best results.

2. It is claimed that using the geometric mean of the cell permeabilities to approximate the flux across volume interfaces is usually not done, but at this point I disagree. Often one uses the geometric or the harmonic mean of the saturated permeabilities and upstream weighting of the relative permeabilities, but to my knowledge it is as common to use geometric means of the total permeability. In textbooks on numerical solutions of two-phase flow equations all these options are usually discussed. It is interesting though, that the geometric mean is a key element to reproduce the fingers.

We thank the reviewer for commenting on this issue. First, continuum models we referred to in the manuscript (see references in lines 56-58) typically use the arithmetic mean for the permeability. However, the paragraph discussing the key role of the geometric mean (lines 292–304) can be understood that the geometric mean is not used at all for the RE. As we already mentioned, the crucial difference between the semi-continuum model and a numerical scheme for the RE is in an appropriate scaling of the retention curve with the block size, not in type of averaging the hydraulic conductivity. Thank you for the correction as the corresponding text in the manuscript is quite misleading. The text will be corrected accordingly.

Second, it is necessary to use a type of averaging that has the desirable property of being small if the permeability of one of the blocks is small. Such an averaging of the hydraulic conductivity creates a pile-up effect, resulting in a finger with saturation overshoot. Thus, the geometric mean is not only possible averaging choice; for example, the harmonic mean can also be used with similar results. In the semi-continuum model, we use the geometric mean because it is shown that using the geometric mean is the most appropriate in our case [11]. The text of the corresponding paragraph (lines 292–304) will be modified and slightly extended to discuss different choices for averaging the hydraulic conductivity.

The effect of appropriate averaging the hydraulic conductivity (for instance using the geometric mean) and scaling of the retention curve can be thus summarised as follows. The geometric mean is essential to create the pile-up effect, while the effect of scaling the retention curve is to preserve this saturation overshoot for $\Delta x \to 0$. This is well observed in Figure 1, where the saturation overshoot disappears for $\Delta x \to 0$ if the scaling of the retention curve is not included, although the geometric mean is used.

3. The authors argue that the model converges to a new type of model if $\Delta x$ goes to zero. If I understand correctly, this goes with the changing capillary pressure saturation curve, which goes from steep to flat with decreasing size $\Delta x$. As the authors write in line 278, the model should be able to reproduce the standard Richards equation behaviour. I do not see how the drainage from a fully saturated soil column towards a hydrostatic profile should be reproducible with this model as $\Delta x$ goes to zero. The hydrostatic saturation

profile should match the primary drainage capillary pressure curve. If the curve gets flat with decreasing $\Delta x$, one would not be able to retrieve the profile. One would get a sharp change from fully saturated to dry.

We first comment on our understanding of line 278. In lines 281–282, we explain that we understand line 278 in the way that the model should be able to reproduce also diffusion-like regime, not only the fingering regime. We do not claim that the semi-continuum model behaves in the same way as the RE in non-overshoot profiles. Lines 281–282 in the manuscript will be corrected to avoid this misunderstanding as follows: "Since the RE can simulate only a diffusion-like regime, we understand (2) in the way that the model should be able to reproduce also diffusion-like regime, not only the fingering regime. This does not mean that the semi-continuum model behaves in the same way as the RE in non-overshoot profiles. This is of course not possible due to the scaling of the retention curve."

Our second comment relates to the reviewer's question of drainage from a fully saturated soil column. We agree with the reviewer that with decreasing $\Delta x$, sharper change from fully saturated to dry is obtained. In Figure 2, simulations of drainage from a fully saturated sand column for three different values of $\Delta x$ are shown at time $t = 15$ minutes. It can be seen that as $\Delta x$ decreases, the saturation profile becomes flatter, so it matches the used draining branch. However, this is not surprising because each block is defined by the same retention curve. This means that the porous medium is perfectly homogeneous, i.e., there is no variance in characteristics of the porous medium. Pražák et al. [12] demonstrated by using model based on percolation theory that in the case of homogeneous network, the retention curve has a step-like form, i.e. it is a constant. This theoretical observation is also consistent with experimental measurements of retention curves [13]. If the pore-size distribution of the porous medium is small, the retention curve tends to be much flatter compared to a situation where the pore size distribution is larger (see figures Fig. 2 and Fig. 3 in [13]). This is consistent with our simulations in which we converge to the constant draining branch.

[Figure]

Figure 2: Simulations of drainage from a fully saturated sand column for three different values of $\Delta x$ at time $t = 15$ minutes.

4. I understand that the authors want to acknowledge the experimental findings of Bauters et al., 2000, and to highlight their achievements by calling the nonmonotonic dependency of finger width with initial saturation Bauters' paradoxon. Still, I find this wording a bit odd. A paradoxon involves a self-contradicting aspect or something that is against the intuitive expectation. This should not diminish the observations, and there is maybe not an easy explanation, but I find it hard to see a paradoxon.

In this case, we respectfully disagree. As the reviewer mentioned, a paradox involves something that is against the intuitive expectation. We believe that the non-monotonic behavior of the finger velocity is unexpected and contrary to common opinion because one would expect the finger velocity to increase with increasing initial saturation. Moreover, the authors of this experiment [10] also describe this behavior as counterintuitive. We provide a quote from the authors below: *"The wetting front velocity was approximately constant in our "observation window" at 15 cm from the top to approximately 15 cm from the bottom. The advance was much slower for the high water contents than for the low water contents. This is **counterintuitive** when classical Richards' type wetting front theory is considered (Fig. 6) that would suggest an increase in velocity with increasing water content."*

Finally, as we mentioned in the manuscript (lines 103–104 and the following text), the experiments of Bauters et al. [10] have almost 90 citations in the Scopus database (to date it is actually more than 90; the number will be changed accordingly in the manuscript), but there is no unified explanation for the observed non-monotonic

behavior. Therefore, we conjecture that this non-monotonic behavior is indeed counter-intuitive because this phenomenon is not sufficiently understood and explained. Hence, the terminology *Bauters' paradox* is used.

5. It is not clear if the model results presented here are predictive or if parameter and other adjustment was involved. In line 245 it is written that a four times lower infiltration rate than in the experiments was used. Why was this lower infiltration rate chosen? Was the match with the experiments not obtained with the same infiltration rate as in the experiments? I find this an important point, as a model needs to be predictive, meaning that one should be able to know the parameters from information about the materials or from measurements of the materials. It is an achievement to reproduce non-monotonic finger width with initial saturation, but if this was obtained with a model that needed fitting, one could argue that one would have obtained the same with a classical (hysteretic) Richards equation model by fitting the capillary pressure curve.

This is an important comment. Again, the presented semi-continuum model is predictive as it is discussed in details in the response to reviewer's comment number 1. The parameters (such as the block size, infiltration rate and material characteristics) are not fitted to obtain the non-monotonic dependence of the finger width and velocity. In addition, we are primarily interested in qualitative agreement with the experiments, not quantitative.

All simulations are computationally demanding. Therefore, we used a lower infiltration rate, which makes the simulation more stable, so that a larger time step $\Delta t$ can be used. For perspective, one simulation presented in the manuscript takes approximately 30 days using distributed computing infrastructure "e-Infrastruktura CZ" (e-INFRA LM2018140). It is important to point out that we did not fit the infiltration rate to reproduce non-monotonic finger width and velocity.

The reviewer RC2 raised a comment regarding the sensitivity analysis. Therefore, we performed the sensitivity analysis and showed that the Bauters' paradox occurs for different values of material parameters or different boundary conditions. In order not to extend this text too much, we refer to our reply to reviewer RC2 (DOI: 10.5194/egusphere-2022-673-AC2) – major issue 2. One part of the sensitivity analysis was the effect of the boundary flux on the flow regime. Five different values of the boundary flux were used ranging from $2 \times 10^{-5}$ ms$^{-1}$ to $32 \times 10^{-5}$ ms$^{-1}$. The non-monotonic dependence of finger width and velocity with initial saturation was observed for all tested values of boundary fluxes i.e., also for $q_B = 32 \times 10^{-5}$ ms$^{-1}$ (see Fig. 4 in the response to the reviewer RC2). Thus, agreement with the experiments (non-monotonic dependence) is obtained at the similar infiltration rate as in the experiments. Let us note that due to the computational demands of the simulations, a larger block size $\Delta x = 0.50$ cm was used for the sensitivity analysis.

Finally, just a little reminder. It has been mathematically proven that in the case of constant boundary flux, the RE is stable regardless of hysteresis. For more details, please see the response to the reviewer's comment number 1.

6. The discussion on the REV in lines 42-51 is a bit long. The problem of the REV for two-phase flow problems has been discussed a lot (for example already in the papers on volume averaging of two-phase flow) and it is acknowledged that the REV for fluid content is problematic, in particular for unstable displacement. The question of an REV for pressure in porous media has also been discussed in the literature (just one example: Nordbotten et al., Water Resources Research 2008). I think this could be shortened and does not need all the citations.

We conjecture that shortening the discussion of REV does not help in understanding the concept of the semi-continuum model. The concept of REV is closely related to the semi-continuum model, specifically the scaling of the retention curve. However, understanding this may not be clear in the present version of the manuscript. This will be explained in more details in the manuscript, as we plan to include a discussion of the semi-continuum model in the manuscript (see our response above the reviewer's comment number 1).

The work of Nordbotten et al. [14, 15] is very interesting. It deals with the role of macroscale pressure and its relation to the pressure in capillaries. However, we decided not to discuss this phenomenon in detail, so the text is not too extensive.

7. Line 56: I think that at least the papers of Lenormand et al., 1983, or Wilkinson, 1986, are here misleading. The point in these papers is not to derive alternative models to the Richards equation but to capture viscous and capillary unstable immiscible displacement. To my knowledge, they do not include gravity.

We agree with the reviewer. The text in the manuscript is a bit misleading. We wanted to stress different approaches to modeling porous media flow, not alternative models to the RE or to capturing finger flow.

Invasion percolation models (thus they do not include external forces such as gravity) are one type for such modeling. However, these two references are indeed misleading, so we will remove them. Instead, we will include a review by Hunt and Sahimi [16]. The text of the corresponding paragraph will be modified for clarity.

8. Line 81: This new type of mathematical models of Nature sounds a bit overselling. The switch from parabolic and hyperbolic for two-phase flow problems is known for continuum models in the limit that capillary effects vanish (also for immiscible displacement in the fractional flow formulation).

We tried to find equations with parabolic-hyperbolic nature caused by a hysteresis operator, but we failed. We were only able to find equations where the special values of a solution give rise to the hyperbolic equation. It usually appears if a respective diffusion coefficient is equal to zero. We also discussed this with our colleague who is interested in hysteresis operators, but without success. We would therefore be indebted to the reviewer for a possible reference related to this kind of equations. In any case, we agree that the used formulation is a bit exaggerated. Hence, this formulation will be modified in the manuscript.

9. Line 237: Where does the 0.0005 come from? It is in none of the figures.

In line 231, it is noted that wetting profiles for $S_{in} = 0.0005, 0.002, 0.04$ are not included in Fig. 4 to make the figure more readable. The finger width and velocity for these values of $S_{in}$ are shown in figures Fig. 5 and Fig. 6. Moreover, all simulation data (including the data set for $S_{in} = 0.0005$) can be downloaded from [17].

10. Line 242-243: I think this is a bit simplifying. The solutions of the Richards equation without hysteresis are stable, so of course this effect is not captured.

First, let us again note that in the case of nondecreasing boundary flux, the solution of the RE is stable regardless of whether hysteresis is included [9]. We already addressed this issue in our response to reviewer's comment number 1, so to avoid repetition, we refer to that response.

Second, we conjecture that the non-monotonic dependence of finger velocity is indeed counter-intuitive. Further details according to this counter-intuitive behavior are given in the reviewer's comment number 4 regarding Bauters' paradox. For clarity, we will change the part *classical Richards' Equation*, which may evoke the Richards' equation without hysteresis, to the more general *classical theory as the Richards' Equation*.

**References**

[1] R. Vodák, T. Fürst, M. Šír, and J. Kmec, "The difference between semi-continuum model and richards' equation for unsaturated porous media flow," *Scient. Rep.*, vol. 12, p. 7650, 2022.

[2] R. G. Larson and N. R. Morrow, "Effects of sample size on capillary pressures in porous media," *Powder Technology*, vol. 30(2), pp. 123–138, 1981.

[3] B. K. Mishra and M. M. Sharma, "Measurement of pore size distributions from capillary pressure curves," *American Institute of Chemical Engineers Journals*, vol. 34(4), pp. 684–687, 1988.

[4] D. Zhou and E. H. Stenby, "Interpretation of capillary-pressure curves using invasion percolation theory," *Transport Porous Med.*, vol. 11, pp. 17–31, 1993.

[5] E. Perfect, L. D. McKay, S. C. Cropper, S. G. Driese, G. Kammerer, and J. H. Dane, "Capillary pressure–saturation relations for saprolite: Scaling with and without correction for column height," *Vadose Zone Journal*, vol. 3(2), pp. 493–501, 2004.

[6] A. G. Hunt, R. P. Ewing, and R. Horton, "What's wrong with soil physics," *Soil Science Society of America Journal*, vol. 77, p. 1877– 1887, 2013.

[7] B. Ghanbarian, V. Taslimitehrani, G. Dong, and Y. A. Pachepsky, "Sample dimensions effect on prediction of soil water retention curve and saturated hydraulic conductivity," *Journal of Hydrology*, vol. 528, pp. 127–137, 2015.

[8] J. A. White, R. I. Borja, and J. T. Fredrich, "Calculating the effective permeability of sandstone with multiscale lattice boltzmann/finite element simulations," *Acta Geotechnica*, vol. 1, pp. 195–209, 2006.

[9] T. Fürst, R. Vodák, M. Šír, and M. Bíl, "On the incompatibility of Richards' equation and finger-like infiltration in unsaturated homogeneous porous media," *Water Resour. Res.*, vol. 45(3), p. W03408, 2009.

[10] T. W. J. Bauters, D. A. DiCarlo, T. Steenhuis, and J.-Y. Parlange, "Soil water content dependent wetting front characteristics in sands," *J. Hydrol.*, vol. 231-232, pp. 244–254, 2000.

[11] J. Jang, G. A. Narsilio, and J. C. Santamarina, "Hydraulic conductivity in spatially varying media - a pore-scale investigation.," *Geophysical J. International*, vol. 184, pp. 1167–1179, 2011.

[12] J. Pražák, M. Šir, and M. Tesař, "Retention cruve of simple capillary networks.," *J. Hydrol. Hydromech.*, vol. 47, pp. 117–131, 1999.

[13] A. Tuli and J. W. Hopmans, "Effect of degree of fluid saturation on transport coefficients in disturbed soils," *European Journal of Soil Science*, vol. 55, no. 1, pp. 147–164.

[14] J. Nordbotten, M. Celia, H. Dahle, and S. Hassanizadeh, "Interpretation of macroscale variables in Darcy's law," *Water Resources Research*, vol. 43, Aug. 2007.

[15] J. Nordbotten, M. Celia, H. Dahle, and S. Hassanizadeh, "On the definition of macroscale pressure for multiphase flow in porous media," *Water Resources Research*, vol. 44, no. 6, 2008.

[16] A. G. Hunt and M. Sahimi, "Flow, transport, and reaction in porous media: Percolation scaling, critical-path analysis, and effective medium approximation.," *Reviews of Geophysics*, vol. 55, p. 993–1078, 2017.

[17] J. Kmec, M. Šír, T. Fürst, and R. Vodák, "Simulation data for: Semi-continuum modelling of unsaturated porous media flow to explain the bauters' paradox, zenodo [data set]," 2022.

---

## Author Response (AR1)

**Author's response for a revision**

In order not to extend this text too much, we do not copy the entire reviewers' comments, but only refer to them. For example, for a comment number one of reviewer RC1 we refer to **RC1_comment1**. Moreover, we do not repeat our answers here because these are already included in point to point responses to reviewers. We changed the manuscript according to the proposed changes in these point to point responses.

In the revised manuscript, the new text is highlighted in blue and the text we removed is highlighted in red. In this document we list all the changes made in the revised manuscript.

**Reviewer RC1**

Our point to point response to reviewer RC1 can be found here:
https://doi.org/10.5194/egusphere-2022-673-AC1
Based on this response, we implemented the following changes in the revised manuscript.

- **RC1_comment1**. Third reviewer's comment (**RC1_comment3**) addresses the same issue, please see below.

- **RC1_comment2**.
  - We replaced the first paragraph as suggested in our response to RC1, hence the proposed text in response to RC1 was copied into the revised manuscript. See lines 9-23 (p. 1–2). The proposed text was slightly modified to avoid repetition in the second paragraph. Moreover, the first sentence from the second paragraph was removed (lines 24–25, p. 2).
  - Other parts of this comment are related to **RC1_comment3**, please see below.

- **RC1_comment3**.
  - As we suggested in response to RC1, we added a new subsection 2.3, in which we discuss a concept of the semi-continuum model and its limit in spatial variable (lines 198–237, p. 7–9). The proposed text in response to RC1 was therefore copied into this new section.
  - The corresponding text referring to the new subsection was added (lines 178–182).
  - The paragraph on REV (lines 51–60, p. 2–3) was slightly changed and moved to subsection 2.3, which describes the role of the REV in more detail.
  - Finally, we stressed that we use the previously developed semi-continuum model to describe the Bauters' paradox (line 94, p. 4).

- **RC1_comment4**. For clarity, we added a new sentence to the description of the Bauters' paradox. See lines 100–101 (p. 4).

**Reviewer RC2**

Our point to point response to reviewer RC2 can be found here:
https://doi.org/10.5194/egusphere-2022-673-AC2
Based on this response, we implemented the following changes in the revised manuscript.

- **RC2_comment1**. Note that this reviewer's comment is similar to **RC1_comment3**. Please, see our changes in the revised manuscript described in **RC1_comment3**.

- **RC2_comment2**. Sensitivity analysis was included in the revised manuscript as we suggested in response to RC2. Some parts of the sensitivity analysis were included in the main part of the revised manuscript and some parts in Appendix B. We copied the proposed text in response to RC2 as follows:

- A part of the sensitivity analysis was included in Appendix B, specifically the effect of intrinsic permeability and dynamic viscosity, relative permeability and retention curve on the flow regime. We decided to include this part in the appendix so that the main part of the revised manuscript would not expand too much. Therefore, a new section Appendix B was created for the purpose of sensitivity analysis. See lines 455–516 (p. 23–28) and corresponding figures Fig. B1–B6 (p. 23–28).

- We believe that the effect of the boundary flux on the flow regime is relevant to the continuity of the manuscript and therefore this part of the sensitivity analysis was added in the main part of the revised manuscript. For this reason, a new subsection 3.5 was created. See lines 319–347 (p. 14–15) and new Figure 8 (p. 15).

- Moreover, one sentence was added to the discussion regarding sensitivity analysis (lines 374–375, p. 16).

- All simulation data related to the sensitivity analysis were uploaded to the Zenodo repository. The reference was changed in the revised manuscript to refer to the new version of the dataset.

- **RC2_comment3**.

  - As suggested in response to RC2, we included simulations without distribution of the intrinsic permeability in Appendix A; see lines 448–454 (p. 20) and Figure A1 (p. 20). The corresponding text in the revised manuscript was changed (line 272, p. 11).

  - Simulation data were uploaded to the Zenodo repository. The reference was changed in the revised manuscript to refer to the new version of the dataset.

- **RC2_comment4**. The proposed text in response to RC2 was copied into the revised manuscript (lines 354–367, p. 16).

- **RC2 minor comments**. Minor issues were fixed.

  - Figures Fig. 5 and Fig. 6 were fixed (p. 13).
  - Units were specified for figures Fig. 3 (p. 11) and Fig. A2 (p. 20).

**Reviewer RC3**

Our point to point response to reviewer RC3 can be found here:
https://doi.org/10.5194/egusphere-2022-673-AC3
Based on this response, we implemented the following changes in the revised manuscript.

- **RC3_comment1**. Many changes were made related to this comment. All the implemented changes were already suggested in response to RC3. For the sake of clarity, we summarise the changes here.

  - A new subsection 2.3 was added, please see **RC1_comment3** for details.
  - We stressed that in the case of boundary influx, the Richards' Equation is unconditionally stable regardless of whether the hysteresis is included (lines 64–65, p. 3).
  - The proposed text in response to RC3 was slightly modified and copied into the revised manuscript (lines 394–399, p. 17).
  - We modified the part of the discussion related to the Richards' Equation (see changes in lines 399–421, p. 17–18). Moreover, the caption of Fig. 9 was changed accordingly (p. 18).

- **RC3_comment2**. The text discussing the role of geometric mean was modified in the revised manuscript to reflect our response to RC3 (lines 399–414, p. 17). This comment is partly related to the previous comment **RC3_comment1**.

- **RC3_comment3**. The proposed text in response to RC3 was copied into the revised manuscript (lines 383–384, p. 17).

- **RC3_comment4**.

  - We stressed in the revised manuscript that the non-monotonic behavior of the finger width and velocity is counterintuitive (lines 44–45, p. 2).

- We also updated the number of citations in the Scopus database for the manuscript Bauters et al. (2000). See lines 116–117 (p. 4).

- **RC3_comment5**.

  - As we mentioned in response to RC3, the sensitivity analysis was performed. Please, see **RC2_comment2** for more details.

  - Moreover, we also added a new paragraph explaining that the semi-continuum model is predictive (lines 394–399, p. 17).

- **RC3_comment6**. This reviewer's comment is already addressed in **RC1_comment3**: The paragraph on REV (lines 51–60, p. 2–3) was slightly changed and moved to subsection 2.3, which describes the role of the REV in more detail.

- **RC3_comment7**. A misleading text in the revised manuscript was changed (lines 66–67, p. 3). Both references (Wilkinson 1986 and Lenormand 1983) were removed (lines 69–70, p. 3) and a new reference (Hunt and Sahimi 2017) was added (lines 77–78, p. 3).

- **RC3_comment8**. We removed the inappropriate sentence on lines 92–93 (p. 4).

- **RC3_comment9**. As we explained in response to RC3, wetting profiles for some values of initial saturation are not included in Fig. 4 to make this figure more readable. For clarity, we stressed this in the revised manuscript (line 287, p. 12). All simulation data can of course be downloaded from Zenodo repository.

- **RC3_comment10**. As suggested in response to RC3, we changed *classical Richards' Equation* to more general *classical theory as the Richards' Equation* (line 299, p. 12).

**Changes not related to reviewers' comments**

- The affiliation of the two authors was slightly modified (p. 1).

- We use the geometric mean of the hydraulic conductivity for computing the flux between neighboring blocks. However, in some cases we wrote about the geometric mean of the relative permeability, which was incorrect. This was corrected in the revised manuscript.

---

## Author Response (AR2)

**Author's response for a revision**

We found two types of typos in the final version of the manuscript, which we corrected.

- Instead of Fig. A2 we referred to Fig. A1. The same applies to Fig. A2–A8. This has been corrected in the manuscript.

- We used the wrong reference for the figure panels. Instead of Fig. 8a, Fig. B1a, etc. we referred to Fig. 8**(a)**, Fig. B1**(a)**. This typo appeared in sections Sensitivity Analysis and Appendix B. This has been corrected in the manuscript.